# Carbon-halogen bond substitution enables high-utilization four-electron iodine redox in noncorrosive dilute electrolytes

Zhiheng Shi[1], Yongchao Tang [1,2] ✉, Yue Wei[3], Guigui Liu[1], Haolong Huang[1], Jintu Qi[1], Zhenfeng Feng[1], Minghui Ye[1,2], Yufei Zhang[1,2], Zhipeng Wen[1,2], Xiaoqing Liu[1,2], Qi Yang [4], Chunyi Zhi [5] ✉ & Cheng Chao Li[1,2] ✉

Aqueous $Zn||I_2$ batteries, involving $I^-/I^0/I^+$ redox, are promising yet usually facing low $I_2$ utilization dominated by $I^0/I^+$ redox, especially under high loadings. Unlocking alternative pathway to $I^0/I^+$ redox, preferably in noncorrosive dilute electrolytes, is a crucial solution. Here, we report a pathway towards more thermodynamically favorable $I^0/I^+$ redox, via a unique carbon-halogen bond substitution. This pathway is realized with a low-concentrated (0.7 M), noncorrosive organohalide additive (2-bromoacetamide, BrAce), triggering a reversible $Br-C\cdots I^{(0)}$ and $C-I^{(+)}-Br$ bond substitution. Compared with conventional interhalogen bonding (I-Br) pathway, this pathway synchronously lowers the barrier for $I^0/I^+$ redox and strengthens the anti-hydrolysis of $I^+$ species, by elaborately regulating axial δ hole activity of interhalogen bond ($I^{(6+)}-Br$). Notably, this pathway enables sustainable operation of four-electron $Zn||I_2$ batteries with high $I_2$ loading ($8.6 \sim 24.0$ mg cm$^{-2}$), featuring improved performances: (1) high $I_2$ utilizations ($55\% \sim 80\%$) at high rates ($5.8 \sim 46.4$ mA cm$^{-2}$), (2) long lifespan (> 400 cycles) with practical areal capacity ($\sim 3.85$ mA h cm$^{-2}$) and 99.5% retention even at 47.5 mA cm$^{-2}$. This pathway opens an exciting research direction to unlock unusual halogen chemistry for scalable, high-energy, sustainable aqueous batteries.

Aqueous $Zn||I_2$ batteries, involving cascaded $I^-/I^0/I^+$ redox, are promising high-energy systems for various potential application scenarios[1,2]. Despite the good compatibility of $I^-/I^0$ redox with conventional electrolytes, $I^0/I^+$ redox works mostly in highly corrosive, halogen anion ($X=Br^-$, $Cl^-$)-concentrated electrolytes[3–6]. The corrosive cell environment shortens the lifespan of the Zn negative electrode due to pitting effects[7,8]. In addition, the use of highly concentrated electrolytes compromises the cost merits of batteries[9,10]. More crucially, the $I^+$ species hydrolysis can accelerate the shuttle effect, leading

to low $I_2$ utilization (below 30%) at high loadings ($\geq 10$ mg cm$^{-2}$)[11]. Hence, the key to promoting the sustainable application of full cells lies in simultaneously triggering $I^0/I^+$ redox with high anti-hydrolysis and prolonging Zn negative electrode lifespan, preferably in mild and dilute electrolytes[12–15].

Implementing $I^0/I^+$ redox involves the formation and stabilization of $I^+$ species, which are usually generated in the form of interhalogen compounds (e.g., IBr, ICl)[3,16–19]. In halogen anion ($X^-=Br^-$, $Cl^-$)-dilute electrolytes, the generated interhalogen compounds are prone to

[1]School of Chemical Engineering and Light Industry, Guangdong University of Technology, Guangzhou, China. [2]Guangdong Provincial Laboratory of Chemistry and Fine Chemical Engineering Jieyang Center, Jieyang, China. [3]School of Environment and Civil Engineering, Dongguan University of Technology, Dongguan, Guangdong, China. [4]State Key Laboratory of Chemical Resource Engineering, College of Chemical Engineering, Beijing University of Chemical Technology, Beijing, China. [5]Department of Materials Science and Engineering, City University of Hong Kong, Kowloon, Hong Kong, China. ✉e-mail: tyc@gdut.edu.cn; cy.zhi@cityu.edu.hk; licc@gdut.edu.cn

hydrolysis, due to the presence of active axial δ hole of interhalogen bonds[20,21]. For the long-term implementation of $I^0/I^+$ redox, a strategy is appropriately weakening the activity of the axial δ hole on the I side ($δ^+ \cdots I$-X). In halogen anion ($X^- = Br^-$, $Cl^-$)-concentrated electrolytes, the generated interhalogen compounds (IX) can further form chain molecule structures (e.g., $[\cdots I\text{-}X\cdots I\text{-}X\cdots]^n$) or anion clusters (e.g., $IX_2^-$) via axial δ hole interaction[4,22–25], which represents the typical interhalogen bonding pathway (Fig. 1a) during $I^0/I^+$ conversion. Therefore, the stabilization of IXs usually needs $X^-$-concentrated chemical environments[6,26]. The key to implementing $I^0/I^+$ redox in mild and dilute electrolytes lies in breaking this shackle, by constructing paradigms to replace the chain IX molecule structures or anion clusters. Although in organic compounds, halogen can stably bond with non-metallic elements (e.g., C, S, N, P)[27,28], no studies have addressed axial δ hole regulation of I-X by non-metallic element-X bonding in organic molecules. Recent studies showed that linking the electron-drawing groups to the I side of IXs, can enhance the halogen bonding constant ($K_a$) by 2-3 orders of magnitude[29]. The introduction of non-interhalogen bonding could be a latent strategy to regulate the axial δ hole of I-X towards more thermodynamically favorable $I^0/I^+$ redox.

Herein, we decouple $I^0/I^+$ redox kinetics from harsh electrolyte environments with a noncorrosive, environment-benign organohalide additive (BrAce, 0.7 M). This additive enables a pathway with a carbon-halogen bond substitution (Br-C$\cdots I^{(0)} \leftrightarrow$ C-$I^{(+)}$-Br) (Fig. 1b) that alters the electronic landscape of iodine intermediates. Compared with the conventional interhalogen bonding pathway, our approach strategically modulates the axial $δ^+$ hole activity of I-X bonds, simultaneously reducing the barrier for $I^0/I^+$ conversion and suppressing hydrolysis of $I^+$ species. This dual regulation enables sustainable operation of four-electron Zn$\|I_2$ batteries with high $I_2$ loading: (1) high $I_2$ utilizations (55% ~ 80%) at high rates (5.8 ~ 46.4 mA cm$^{-2}$), (2) long lifespan ( > 400 cycles) with areal-specific capacity ( ~ 3.85 mA h cm$^{-2}$) and 99.5% retention at a high current density (47.5 mA cm$^{-2}$). This pathway eliminates reliance on corrosive halide salts, prolonging Zn negative electrode lifespan while reducing electrolyte toxicity and cost. Beyond iodine, this carbon-halogen substitution concept offers a universal design principle for stabilizing multivalent halogen redox towards scalable, high-energy, and sustainable aqueous batteries.

## Results

### $I^0/I^+$ redox triggered in noncorrosive dilute electrolytes

Among interhalogen compounds (XY), the bigger the electronegativity differences (△EN) between X and Y, the more stable the XY is, but the more difficult it is to form bonds[30]. Based on △EN values for IBr (0.3), ICl (0.5), and IF (1.5), $I^+$ species should be more challenging to be triggered by additives with C-Cl and C-F bonds[31]. We focus on the $I^0/I^+$ redox triggered by organic additives with C-Br bonds. Balancing cost and ionic conductivity, a sulfate electrolyte with 3 M ZnSO$_4$ formulation is employed as baseline (denoted ZSO)[32]. Water-soluble organic additives, with a formation of R-CH$_2$-Br (R = electron-donating or -withdrawing groups) (Fig. 2a), are respectively introduced into ZSO to examine their efficacy. For comparison, ZnBr$_2$ salts are introduced into ZSO as well.

The regulation efficacy of additives is scrutinized by cyclic voltammetry (CV) tests of Zn$\|I_2$ batteries ($I_2$ loading: 3 mg cm$^{-2}$; cut-off voltage: 0.3 − 1.75 V; scan rate: 1 mV s$^{-1}$) (Fig. 2b). Acetamide (Ace) without C-Br bonds and 2-bromomethyl tetrahydrofuran (BrTHF) with an electron-donating group cannot activate $I^0/I^+$ redox (the pair of peaks at higher potential), whereas BrAce, bromoacetone (BrPK), 3-bromopropionamide (3-BrPace), and bromoacetonitrile (BrAN) with electron-withdrawing groups show distinct $I^0/I^+$ redox behavior. The corresponding Galvanostatic charge-discharge (GCD) curves (Fig. 2c) at 1 A g$^{-1}$ further confirm this trend, i.e., no plateaus of $I^0/I^+$ redox for Ace and BrTHF (Supplementary Figs. S1 and S2). Still, they appear with BrAce, BrPK, 3-BrPace, and BrAN (Supplementary Figs. S3–S5). These results verify that the activation of $I^0/I^+$ redox is highly associated with the C-Br bond and the electronic properties of R. When C-Br bonds and electron-withdrawing R coexist, the electron-drawing ability (Hammett parameter: σ > 0; Supplementary Table S1) of R influences $I_2$ utilization (ratio of theoretical capacity to actual capacity of $I^-/I^0/I^+$ redox) (Fig. 2d). The σ value can be used to quantify the electron-withdrawing strength of the organic additives, i.e., a more positive σ value indicates greater electron-withdrawing ability[33]. Compared to BrAce with amide group (σ = 0.28), BrAN (nitrile, σ = 0.66) and BrPK (ketone, σ = 0.51) additives provide inferior $I_2$ utilization. Theoretically, optimal electron-withdrawing groups can effectively bind to $I^+$ species, preventing them from acting as independent active sites. This consequently weakens the nucleophilic attack of H$_2$O molecules and thus enhances the cycling stability of the battery. However, an excessive electron-withdrawing effect reduces the nucleophilicity of the C-Br bond, hindering the formation of I-Br. In contrast, electron-donating groups compromise the stability of charged products and accelerate capacity fading. The amide group in BrAce shows a slightly higher σ value than that of Br (0.28 vs. 0.23). The slightly higher σ value of electron-withdrawing groups than that of bromine of C-Br bond could be more beneficial to achieve the trade-off for iodine redox reversibility and $I_2$

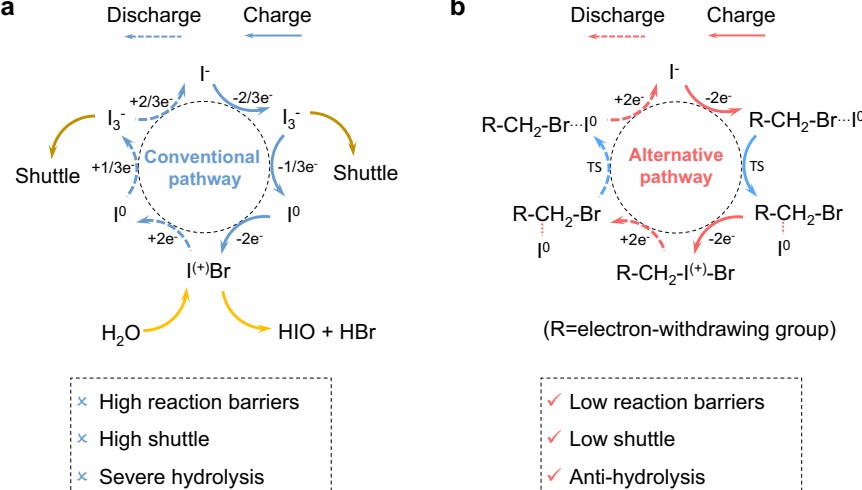

**Fig. 1 | Reaction pathways to I0/I+ redox in four-electron Zn||I2 batteries. a** Conventional pathway by interhalogen bonding. **b** Alternative pathway by carbon-halogen bond substitution.

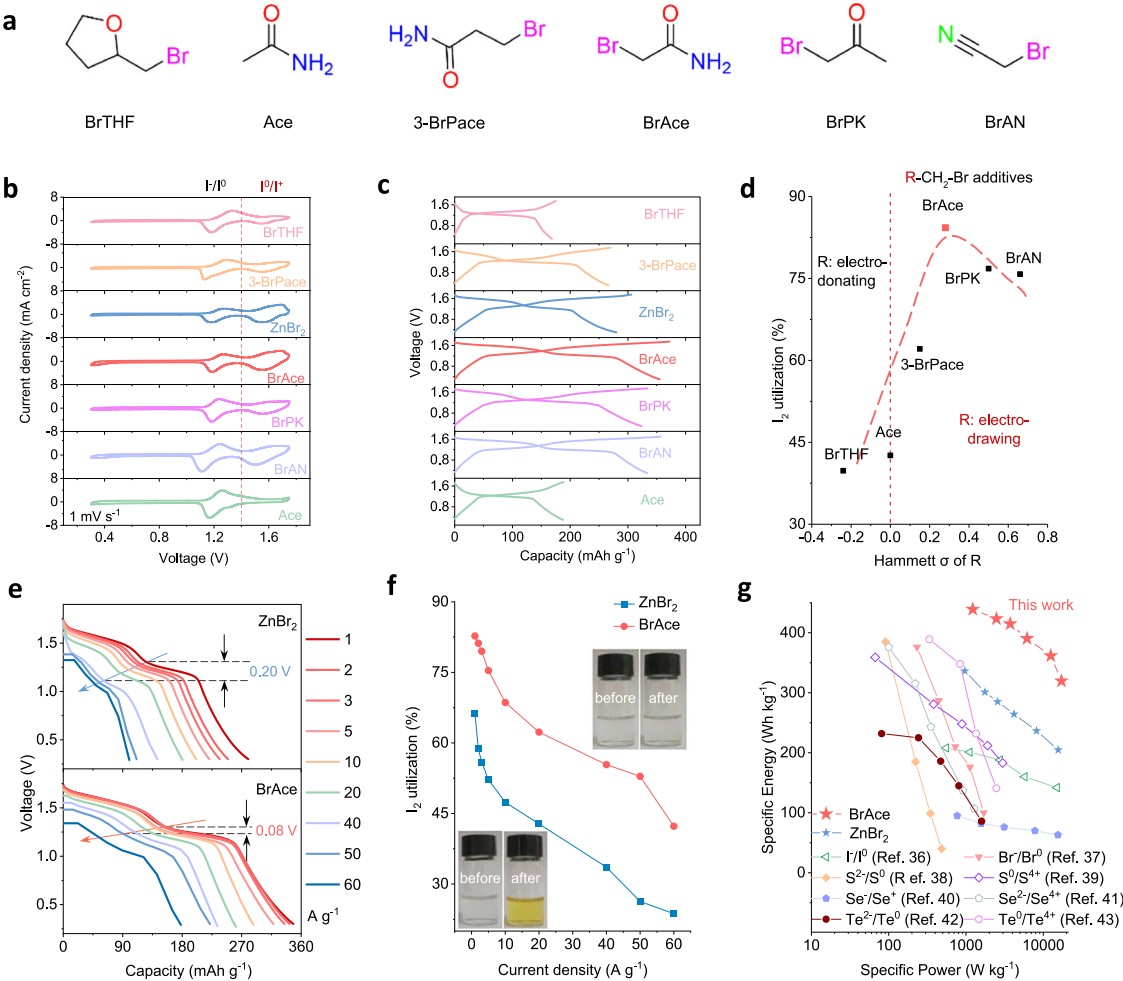

**Fig. 2 | Four-electron aqueous Zn||I2 batteries with different additives as activators. a** Molecular structures of organic additives with different R groups. **b** CV curves of Zn||I$_2$ batteries (I$_2$ loading: 3 mg cm$^{-2}$; cut-off voltage: 0.3 - 1.75 V) in electrolytes with different additives at 1 mV s$^{-1}$. **c** The corresponding GCD curves at 1 A g$^{-1}$. **d** Correlation of I$_2$ utilization with σ values of R in organic additives. **e, f** Discharge curves of Zn||I$_2$ batteries at high rates of 1-60 A g$^{-1}$ and the corresponding distribution of I$_2$ utilizations (Inset: BrAce and ZnBr$_2$ electrolytes before and after cycling). **g** Ragone plot comparison of Zn||I$_2$ batteries with other conversion-type systems.

utilization. Among six additives, BrAce with amide group (σ = 0.28) allows for the highest I$_2$ utilization (83.1%), confirming the crucial role of the appropriate σ value in triggering the I$^0$/I$^+$ redox. Hence, the σ value could be an effective descriptor for regulation of I$^0$/I$^+$ redox triggered by C-Br bonds. Besides, with chloroacetamide (ClAce) and 2,2-difluoroacetamide (DFAce) additives, I$^0$/I$^+$ redox can be triggered only at higher voltage windows ( ~1.85 V for ClAce, 1.95 V for DFAce), and Zn||I$_2$ batteries typically show inferior cycling stability (Supplementary Figs. S6 and S7). This result conforms to the higher △$EN$ values for ICl (0.5), and IF (1.5).

BrAce is selected for subsequent electrochemical investigation. To accurately recognize the effect of C-Br bonds, ZnBr$_2$ salts with equimolar Br$^-$ are added into ZSO for comparison. The introduction of BrAce enhances the overpotential of hydrogen and oxygen evolution reactions, as revealed by linear sweep voltammetry (LSV) curves (Supplementary Fig. S8). Compared to other high-concentration salt electrolytes, the BrAce electrolyte exhibits lower viscosity and higher ionic conductivity, which is beneficial to obtain four-electron Zn||I$_2$ batteries with fast redox kinetics (Supplementary Fig. S9). When the Zn||I$_2$ battery is further charged to 1.85 V at 1 A g$^{-1}$, a higher charge plateau appears at 1.78 V (Supplementary Fig. S10), corresponding to the Br$^-$/Br$^0$ conversion; during discharging, the corresponding discharge plateau appears, along with a decrease in the Coulombic

efficiency (CE). When scanned at 1 mV s$^{-1}$ up to 2.0 V, CV curves of Zn||I$_2$ batteries also show the Br$^0$/Br$^+$ redox peaks (Supplementary Fig. S11)[34]. It indicates that excessively high charging voltage causes electrolyte decomposition, leading to the irreversible fracture of C-Br bonds of additives. Furthermore, Zn battery using iodine-free positive electrode is evaluated in the BrAce electrolyte (specific current: 1 A g$^{-1}$; carbon loading: 2.5 mg cm$^{-2}$; voltage window: 0.3 - 1.75 V). The GCD curves (Supplementary Fig. S12) display a typical capacitive behavior, without identified bromine redox conversion. It suggests that the cut-off voltage of 0.3 - 1.75 V can eliminate bromine redox interference in four-electron iodine conversion. To ensure the efficacy of BrAce, an upper voltage limit of 1.75 V is chosen for the following battery tests.

The solubility limit of BrAce in ZSO is approximately 1.0 M. Among various BrAce concentrations, 0.7 M BrAce endows the Zn||I$_2$ battery with the highest capacity and the longest cycling stability at 1 A g$^{-1}$ (Supplementary Fig. S13). The near-saturated BrAce can reduce the reversibility of iodine conversion to some extent. Therefore, 0.7 M BrAce and 0.35 M ZnBr$_2$ were introduced into ZSO as electrolytes, respectively. At high rates of 1-60 A g$^{-1}$, discharge curves of Zn||I$_2$ batteries show the advantage of BrAce over that by ZnBr$_2$, in triggering I$^-$/I$^0$/I$^+$ redox with higher capacity and well-kept plateaus (Fig. 2e). When the specific current increases from 1 to 50 A g$^{-1}$, Zn||I$_2$ batteries in BrAce electrolytes display much smaller polarization than that in ZnBr$_2$

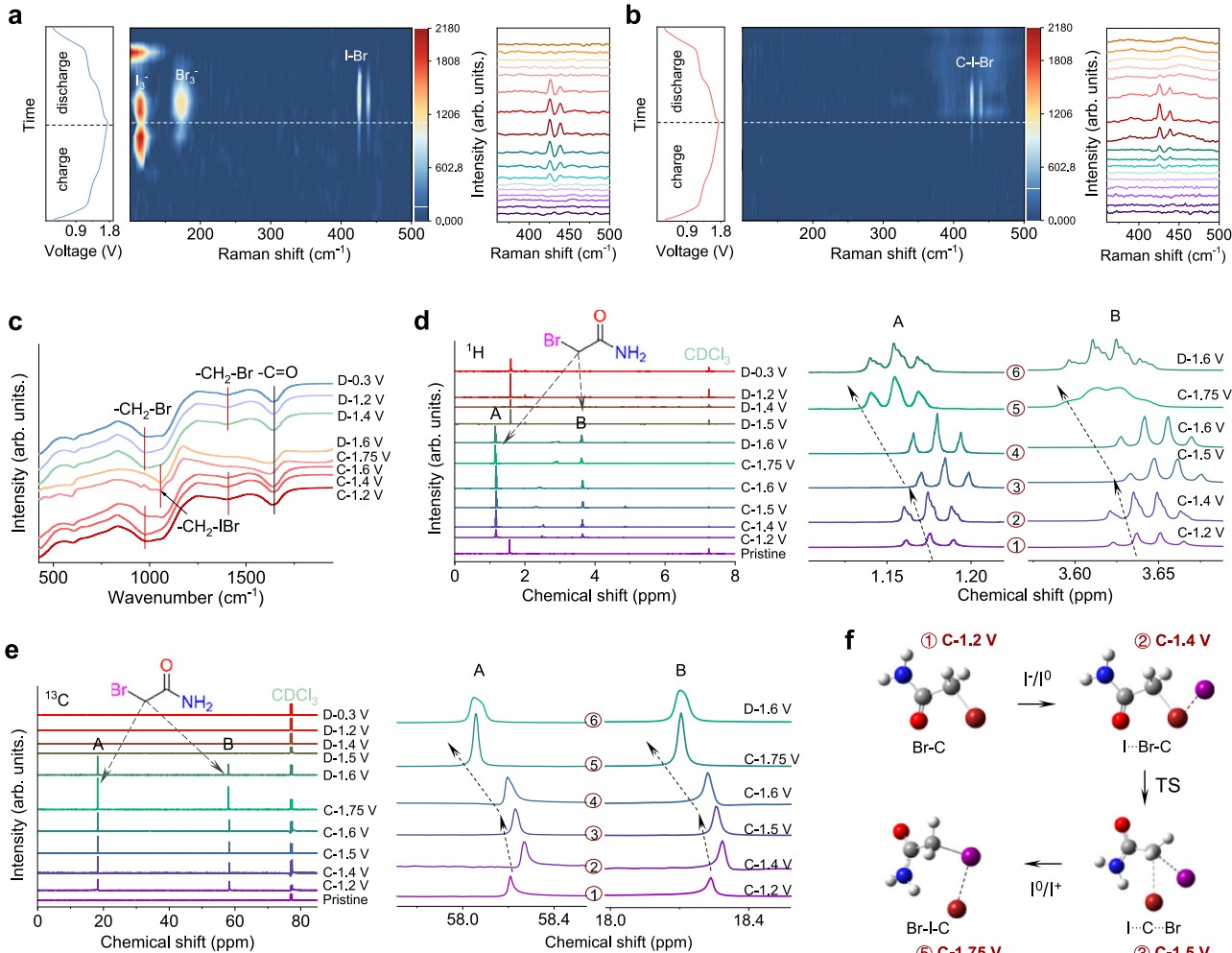

**Fig. 3 | Carbon-halogen bond substitution pathway to I0/I+ redox. a, b** In situ Raman spectra of positive electrodes during GCD in 0.35 M ZnBr$_2$ and 0.7 M BrAce electrolytes (cycling current: 0.5 A g$^{-1}$). **c** In situ ATR-IR spectra of positive electrodes during GCD in 0.7 M BrAce electrolytes (cycling current: 0.5 A g$^{-1}$). **d** Ex situ $^1$H NMR spectra of positive electrodes during GCD in 0.7 M BrAce electrolytes and magnified $^1$H NMR spectra corresponding to peak A and peak B. **e** Ex situ $^{13}$C NMR spectra of positive electrodes during GCD in 0.7 M BrAce electrolytes and magnified $^{13}$C NMR spectra corresponding to peak A and peak B (After 3 cycles at a current density of 0.5 A g$^{-1}$, the iodine electrodes at different states of charge were disassembled and collected for subsequent characterizations). **f** Schematic illustration of carbon-halogen bonding substitution pathway to I0/I+ redox. Red spheres: O atoms; Blue spheres: N atoms; Brown spheres: Br atoms; Purple spheres: I atoms; Gray spheres: C atoms; White spheres: H atoms. TS: transition state.

electrolytes (0.08 V vs. 0.20 V). The trend is more obvious when discharge curves at different rates are compared individually (Supplementary Fig. S14). The Zn‖I$_2$ batteries with BrAce also show higher I$_2$ utilization than that with ZnBr$_2$ (Fig. 2f). Even at 50 A g$^{-1}$, also > 50% I$_2$ utilization remains obtained with BrAce, while the counterpart with ZnBr$_2$ is only ~25%. Color changes in the cycled BrAce electrolytes (always colorless) and ZnBr$_2$ electrolytes (from colorless to shallow yellow) manifest no Br$_3^-$ and a large number of Br$_3^-$ formation (inset in Fig. 2f), respectively[35]. This high I$_2$ utilization renders the Zn‖I$_2$ batteries with merits of specific energy/power (~ 450 Wh kg$^{-1}$$_{Iodine}$, -18 kW kg$^{-1}$$_{Iodine}$) over other conversion positive electrode chemistries (Fig. 2g, Supplementary Table S2), including I$^-$/I$^0$, Br$^-$/Br$^0$, and multi-electron chalcogen redox[36–43].

## Carbon-halogen bond substitution pathway to I0/I+ redox

To unveil the BrAce-regulated pathway to I0/I+ redox, in situ Raman spectroscopy is initially conducted to track iodine speciation dynamics on the positive electrode side during GCD cycling (Fig. 3a, b). The Zn‖I$_2$ batteries are firstly discharged to 0.3 V (D-0.3 V) and then scrutinized by Raman spectroscopy. The conventional interhalogen compounds

(i.e., IBr) form when Zn‖I$_2$ batteries operated in Br$^-$-rich electrolytes in previous studies[5]. Raman spectra in Fig. 3a show the presence of the IBr signal (from charged to 1.6 V (C-1.6 V) to discharged to 1.6 V (D-1.6 V), at 428 cm$^{-1}$), suggesting the conventional pathway to I0/I+ redox via interhalogen bonding (I-Br) in 0.35 M ZnBr$_2$ electrolytes[17,34]. Meanwhile, when charged to 1.4 V (C-1.4 V), I$_3^-$ species appear (at 110 cm$^{-1}$), which usually results in shuttle effect[44]. Additionally, during IBr formation, strong signals assigned to Br$_3^-$ species appear at 160 cm$^{-1}$, indicating the coexistence of IBr formation and hydrolysis[19,35,45]. The hydrolysis dissociates Br$^-$ to initiate Br$^0$/Br$_3^-$ redox, which would accelerate negative electrode corrosion and lower conversion kinetics, leading to battery decay. In contrast, I-Br signals in 0.7 M BrAce electrolytes are well-kept from C-1.6 V to D-1.6 V (Fig. 3b), demonstrating reversible I0/I+ conversion. Notably, no polyiodide (I$_3^-$) or polybromide (Br$_3^-$) signals are detected, suggesting that the BrAce-regulated pathway effectively suppresses hydrolysis and prompts cascaded I$^-$/I$^0$/I$^+$ redox. Additionally, in 0.7 M BrAce electrolytes, I-Br signals appear a red shift (Supplementary Fig. S15), compared to the counterparts in 0.35 M ZnBr$_2$ electrolytes. This results from the electron-withdrawing effect of amide group. Notably, the in situ Raman spectra exhibit

weakened peak intensity for C-I·Br compared to I-Br (Fig. 3a, b). This is attributed to that the amide group in BrAce reduces the electron density of adjacent atoms or functional groups, thereby lowering the polarizability of charged intermediates[46]. These results confirm that the I-Br bond in the BrAce system is not isolated like conventional one, but connected by electron-withdrawing organic group.

The intermediates at different charge/discharge potentials are further characterized by X-ray photoelectron spectroscopy (XPS). In 0.7 M BrAce electrolytes, both the Br $3d$ and I $3d$ XPS spectra of intermediates at high potentials (C-1.75 V, D-1.6 V) appear the signals associated with $I^+$ species at 74.0 and 631.5 eV (Supplementary Fig. S16), respectively. Both the binding energies are different from those for the counterparts (70.3 eV and 631.8 eV) in 0.35 M $ZnBr_2$ electrolytes (Supplementary Figs. S17 and S18). Specifically, $I^+$ species signals in Br 3 d of intermediates in 0.7 M BrAce electrolytes show a 3.7-eV higher binding energy than those in 0.35 M $ZnBr_2$ electrolytes. Such a large gap manifests that, in addition to bonding between Br and I, the electron density around the Br also reduces, which could be caused by inductive effect of electron-withdrawing amide group. This is consistent with the result revealed by in situ Raman spectra in Fig. 3a-b, i.e., the C-I·Br pathway lowers the polarizability of charged product. In the I $3d$ XPS spectra, the characteristic peak of $I^+$ species is closer to the C-I bond (631.4 eV), indicating similar chemical properties of the $I^+$ species to alkyl iodide substances.

To further discern the C-Br bond evolution of BrAce, the positive electrode during GCD processes is characterized by in situ attenuated total reflectance-infrared spectroscopy (ATR-IR) and ex situ $^1H/^{13}C$ nuclear magnetic resonance (NMR)[47]. ATR-IR spectra in Fig. 3c reveal that, during GCD processes, the signals associated with -C-Br (at 600.6 cm$^{-1}$), -$CH_2$-Br (at 977.9 and 1411.4 cm$^{-1}$, respectively), and -C = O (at 1650.9 cm$^{-1}$) remain, suggesting that BrAce is not electrochemically decomposed[48,49]. However, when charged to 1.75 V (C-1.75 V) and discharged to 1.6 V (D-1.6 V), a new peak appears at 1055.5 cm$^{-1}$, and meanwhile, the peak at 1411.4 cm$^{-1}$ disappears, which could be associated with the isomerization of -$CH_2$-Br caused by the formation of -$CH_2$-IBr. Furthermore, $^1H$ NMR and $^{13}C$ NMR spectra at high potentials (from C-1.2 V to C-1.75 V and to D-1.6 V) in Fig. 3d, e appear split peaks of H (A: ~1.18 ppm; B: ~3.65 ppm) and C (A: ~18.23 ppm; B: ~50.09 ppm) of -$CH_2$-Br, which is different from that of pristine BrAce (Supplementary Fig. S19a). For pristine BrAce, the H in -$CH_2$-Br is under chemical equivalence, thereby showing no splitting. The splitting of H signals in -$CH_2$-Br implies that the chemical equivalence is broken by isomerization of -$CH_2$-Br. To simulate operational environment of BrAce in aqueous batteries, BrAce is dissolved into deionized water for $^1H$ NMR characterization. The amide proton signals (2.45 ppm and 3.3 ppm) almost disappear (Supplementary Fig. S19b-d) due to the interference of hydrogen-bonding interaction between water and amide group. In contrast, the hydrophobic -$CH_2$Br protons remain distinct at 3.8 ppm, demonstrating their relative insensitivity to aqueous solvation. The positive electrode samples during GCD cycling are subsequently analyzed. When charging beyond 1.2 V, a new resonance at 1.18 ppm appears (Fig. 3d). These peaks experience analogous potential-dependent shifts to the -$CH_2$Br signal at 3.65 ppm (-$CH_2$Br proton) (Fig. 3d). This emergent signal is assigned to one proton of the -$CH_2$Br group, where asymmetric electronic environments resulting from interaction (chemical adsorption or bonding) with iodine species decouple the methylene protons, yielding distinct chemical shifts. Such decoupling implies covalent or coordinative bonding between BrAce and iodine intermediates during charge transfer. The splitting phenomena of H and C also accord with the ATR-IR results.

The peak upshift and downshift of $^1H$ NMR peaks are usually caused by adjacent electron-donating and -withdrawing groups, respectively[50]. For $^{13}C$ NMR spectra, the effects of adjacent electron-donating and -withdrawing groups on peak upshift and downshift are consistent with those in $^1H$ NMR spectra[51]. As depicted in Fig. 3d-e, the

electrochemical evolution during charging from 1.2V to 1.4V (stage ①②) initiates the $I^-/I^0$ conversion, where iodine species ($I^0$) function as Lewis acids that initially coordinate with the bromine atom of the -$CH_2$-Br moiety. Given the high polarizability of $I^0$, this Br···I coordination could facilitate electron withdrawal from the Br-C bond through n→σ* anti-bonding orbital interactions. Consequently, the methylene group experiences deshielding, accounting for the downfield shifts in both $^1H$ and $^{13}C$ NMR spectra. Subsequently, during the voltage elevated to 1.5V (stage ②③), the -$CH_2$-Br···$I^0$ undergoes isomerization to form $I^0$···$CH_2$···Br structure. In this rearranged state, $I^0$ transitions into an electron donor, cooperatively enhancing electron density at the methylene carbon alongside bromine. This electronic redistribution elevates shielding effects, thereby inducing upfield shifts of the methylene proton and carbon resonances in NMR spectra. Electrostatic potential simulations further corroborate this dynamic behavior (Supplementary Fig. S20, Supplementary Data 1–3). When the $CH_2Br$···$I^0$ intermediate forms, iodine accumulates electron density while depleting electron density around the methylene group, rationalizing the initial downfield NMR shifts. Following isomerization to the $I^0$···$(CH_2)$···Br intermediate, $I^0$ donates electron density to the methylene carbon, increasing its electron cloud density and inducing the upfield shift. These computational results align with Fig. 3d, e.

We further characterized the charged intermediates (C-1.75 V) in 0.7 M BrAce electrolytes by liquid chromatography-mass spectrometry (LC-MS). In the LC-MS spectra, a strong signal appears at m/z 264.8 (Supplementary Fig. S21), corresponding to the weight of $NH_2$-C(O)-$CH_2$-(IBr), confirming the presence of C-(I·Br) moiety. To identify the bonding mode of C-(I·Br), we investigated the vibration change of C-Br bond in the BrAce system. Based on the Raman spectra of different electrolytes (Supplementary Fig. S22), the characteristic vibration signals of $SO_4^{2-}$ and C-Br bond were identified (at 600 cm$^{-1}$ and 700 cm$^{-1}$, respectively). As shown in Supplementary Fig. S23, within the in situ Raman testing range of 500–800 cm$^{-1}$, only the $SO_4^{2-}$ signals were detected during charge/discharge in the $ZnBr_2$ system. In contrast, within the range of C-1.6 V to D-1.6 V, the BrAce system exhibited reversible split double peaks at around 630 cm$^{-1}$, which is assigned to stretching vibration of C-(I·Br) with I-Br coupling effect. Meanwhile, the characteristic peak of the C-Br bond disappeared with the formation of C-(I·Br) and then gradually recovered after the disappearance of C-(I·Br) (from D-1.6 V to D-0.3 V). This confirms a reversible transformation between the C-Br bond and C-(I·Br). Based on the charge distribution of covalent bonds, the bonding of $I^+$ enhances the covalent bond, leading to an increase in its force constant and a subsequent rise in vibration frequency (manifested as blueshift). This confirms that the C is connected with $I^+$ in the C-(I·Br) moiety, i.e., its presence in the form of C-$I^+$·Br. If bonding occurs in the form of C-Br-$I^+$, the characteristic peak of the C-Br bond should undergo a certain degree of blueshift rather than disappearance after $I^+$ bonding to the end of the C-Br bond. This phenomenon suggests that the C-Br bond breaks and re-bonds in the form of C-$I^+$-Br.

To investigate the chemical properties of C-I·Br, we further compared the $^1H$ NMR of C-I·Br samples (C-1.75 V) with commercial IAce and BrAce. As shown in Supplementary Fig. S24, the characteristic signal peak of the α-hydrogen attached to IBr in C-I·Br appeared between the characteristic peaks of IAce and BrAce. Notably, this signal peak is closer to that of IAce and farther from that of BrAce. It confirms similar chemical properties of the C-I·Br sample to IAce, and provides evidence that the α-carbon is bonded with I rather than Br. During the discharge process, R-$CH_2$–I–Br is reduced to R-$CH_2$Br and $I_2$ ($I^+/I^0$ conversion), followed by the conversion of $I_2$ to $I^-$ at the $I^0/I^-$ stage. The reduced involvement of C-Br bonds in the discharge process accounts for the spectral differences between the charged states (C-1.4 V, C-1.2 V) and discharged states (D-1.4 V, D-1.2 V). The retention of the -$CH_2$-I–Br signal at the high-voltage plateau (C-1.75 V/D-1.6 V) and its complete recovery to the original

-CH$_2$Br signal upon discharge (D-1.4 V/D-1.2 V) confirm the electrochemical reversibility of BrAce.

We further conducted iodine color reactions to verify the reversible transformation between the C-Br bond and C-I-Br, chemical differences between C-I-Br and I-Br, and similarity of chemical properties of C-I-Br to C-I bond in the IAce. Usually, non-polar iodides change color in deuterated chloroform (CDCl$_3$), which essentially arises from the interaction between the low polarity of CDCl$_3$ and iodides via van der Waals forces, and this weak interaction barely alters the electronic structure of iodides[52]. As shown in Supplementary Fig. S25a, BrAce and IAce appeared colorless and magenta in CDCl$_3$, respectively; while I$^-$ (from ZnI$_2$), I$_2$, and I-Br appeared colorless, purple-black, and yellow-brown in CDCl$_3$, respectively. When the products of the iodine positive electrode at different potentials (with the same iodine content) were dissolved in CDCl$_3$, it was found that Br-C···I$^0$ (C-1.3 V) shows the same color as BrAce, while the colors of I···C···Br (C-1.5 V) and C-I-Br (C-1.75 V, D-1.6 V) gradually approached that of IAce (Supplementary Fig. S25b). Finally, we assembled and tested a Zn||I$_2$ battery using IAce (0.4 M, ~5 mg$_I$ cm$^{-2}$) as the iodine source, ZnBr$_2$ (0.35 M) as the initiator, and carbon cloth as the positive electrode current collector. With the cutoff voltage of 0.3 − 1.75 V, distinct I$^-$/I$^0$/I$^+$ redox couple characteristic peaks appeared (Supplementary Fig. S26). In addition, the battery showed no obvious capacity decay during 500 cycles, indicating a stable C-I-Br structure formed in the system as well. This result further verifies the reliability of the above conclusions.

Based on the discussion above, we summarize the carbon-halogen bonding substitution pathway to I$^-$/I$^0$/I$^+$ redox as follows (Fig. 3f, Supplementary Data 1-4):

$$(I^-/I^0): 2I^- - 2e^- + R - CH_2 - Br(BrAce) \rightarrow 2R - CH_2 - Br \cdots I^{(0)}(R, NH_2 - (C=O)-) \tag{1}$$

$$(\text{isomerization transition}): 2R - CH_2 - Br \cdots I(0) \rightarrow 2R - (I^0 \cdots (CH^2) \cdots Br) \tag{2}$$

$$(I^0/I+): 2R - (I^0 \cdots (CH_2) \cdots Br) - 2e^- \rightarrow 2R - CH_2 - I^{(+)} - Br \tag{3}$$

$$(I^+ + /I^0): 2R - CH_2 - I^{(+)} - Br + 2e^- \rightarrow 2R - CH_2Br + I_2 \tag{4}$$

$$(I^0/I^-): I_2 + 2e^- \rightarrow 2I^- \tag{5}$$

**Lowered barrier of I$^-$/I$^0$/I$^+$ redox and anti-hydrolysis of I$^+$ species**
To investigate the redox kinetics and anti-hydrolysis of the carbon-halogen bond substitution pathway, energy barriers are further calculated by density functional theory (DFT). On the basis of the C-Br bond-regulated pathway summarized in Fig. 3f, structural models of species involved in I$^-$/I$^0$/I$^+$ redox are firstly established and optimized. Subsequently, Gibbs free energy changes ($\Delta G$) across all electrochemical steps are calculated (Fig. 4a, Supplementary Table S3). For comparison, $\Delta G$ values for I$^-$/I$^0$/I$^+$ redox regulated by ZnBr$_2$ are also calculated. In contrast to the ZnBr$_2$-regulated pathway (blue), C-Br bond-regulated pathway (red) shows lowered barriers for I$^-$/I$^0$ redox ($\Delta G_{1(red)}$ vs. $\Delta G_{2(blue)}$) and I$^0$/I$^+$ redox ($\Delta G_3$)[13,43]. This result verifies that C-Br bond substitution is far more thermodynamically stable than conventional interhalogen bonding regulation. This is strongly supported by the enhanced rate capability by BrAce (Fig. 2e).

NAt scan rates of 1–20 mV s$^{-1}$, Zn||I$_2$ batteries with BrAce electrolytes exhibit well-defined redox pairs (I$^-$/I$^0$ and I$^0$/I$^+$) (Supplementary Fig. S27). Even at 20 mV s$^{-1}$, minimal peak separation (<120 mV) remains, confirming low polarization and fast kinetics enabled by

BrAce; while Zn||I$_2$ batteries with ZnBr$_2$ electrolyte show widened redox peak separations, indicating severe polarization and sluggish kinetics[53]. Pseudocapacitive contributions in the I$^-$/I$^0$/I$^+$ redox related to redox kinetics are further evaluated by power-law relationship ($i = av^b$, $i$: peak current; $v$: scan rate; $a$ and $b$: constant) (Supplementary Fig. S28)[53,54]. The higher $b$-values feature the kinetics merits of both I$^-$/I$^0$ and I$^0$/I$^+$ redox in BrAce over ZnBr$_2$ electrolytes. Deconvolution of pseudocapacitive ($k_1v$) and diffusion ($k_2v^{1/2}$) contributions at difference scan rates (1-20 mV s$^{-1}$) is further conducted (Supplementary Fig. S29)[55]. Zn||I$_2$ batteries in BrAce electrolytes exhibit higher pseudocapacitive contributions (43%~80%) at various scan rates than those in ZnBr$_2$ electrolytes (27%~69%). Typically, CV curves at 1 mV s$^{-1}$ present the pseudocapacitive contribution distinction (47% vs. 30%) of I$^-$/I$^0$/I$^+$ redox in BrAce and ZnBr$_2$ electrolytes (Supplementary Fig. S30). These further validate enhanced I$^-$/I$^0$/I$^+$ redox kinetics by BrAce relative to ZnBr$_2$.

In addition, based on electrochemical impedance spectra (EIS) at various temperatures (Supplementary Figs. S31–S33) and the Arrhenius equation (Supplementary Fig. S34), we calculate the activation energy ($E_a$) for I$^-$/I$^0$ and I$^0$/I$^+$ redox, respectively (Fig. 4b)[5]. Similar to the results calculated by DFT, both I$^-$/I$^0$ and I$^0$/I$^+$ redox by BrAce show reduced $E_a$ values compared with that by ZnBr$_2$. Specifically, in the case of BrAce, 11.48 kJ mol$^{-1}$ (charge) and 8.83 kJ mol$^{-1}$ (discharge) for the I$^-$/I$^0$ conversion, which is 76%–78% lower than that of ZnBr$_2$ system (56.73 and 33.52 kJ mol$^{-1}$, respectively); while for I$^0$/I$^+$ redox conversion, $E_a$ values are 11.36 kJ mol$^{-1}$ (charge) and 10.40 kJ mol$^{-1}$ (discharge) by BrAce, which is 39%–42% lower than that by ZnBr$_2$ (19.68 and 17.1 kJ mol$^{-1}$, respectively). Such results verify the kinetics advantage of the carbon-halogen bond substitution pathway.

Anti-hydrolysis of I$^+$ species regulated by carbon-halogen bond substitution and interhalogen bonding is further investigated by DFT. To judge the influence of SO$_4^{2-}$ concentration on the stability of R-CH$_2$-IBr, we tested Zn||I$_2$ batteries using 1 M ZnSO$_4$ + 0.7 M BrAce and 2 M ZnSO$_4$ + 0.7 M BrAce as the electrolytes, respectively. These batteries both exhibit no significant capacity decay over 1000 cycles at 2 A g$^{-1}$, with specific capacities of 340~350 mAh g$^{-1}$ following an initial activation (Supplementary Fig. S35). Both iodine utilization and cycling stability are close to the 3 M ZnSO$_4$ + 0.7 M BrAce. These results confirm that I$^0$/I$^+$ redox process is primarily governed by the modulation effect of carbon-halogen bond substitution rather than SO$_4^{2-}$ concentration. Therefore, during the DFT calculations of hydrolysis energy barrier for the I$^0$/I$^+$ redox reaction, the influence of ZnSO$_4$ is rationally neglected. For IBr hydrolysis (Fig. 4c, top), water molecules tend to first attack the active axial δ hole, and then oxygen (electronegative) and hydrogen (electropositive) of H$_2$O combine with I$^{(+)}$ and Br$^{(-)}$ of IBr respectively. Finally, IBr is split into HIO and HBr (Supplementary Data 5–8). This pathway has been mentioned in previous studies[13]. While for R-CH$_2$-IBr hydrolysis (Fig. 4c, bottom), it is hypothesized with a similar pathway to the former, i.e., oxygen and hydrogen of H$_2$O firstly combine with I$^{(+)}$ and Br$^{(-)}$ of IBr, respectively. The final species are IAce with -CH$_2$-I, OH$^-$, and HBr (Supplementary Data 8–11). After structural model optimization, the IBr and R-CH$_2$-IBr hydrolysis barriers are calculated (by subtracting the free energy of the initial reactants from the free energy of the hydrolysis products of R-CH$_2$-IBr or IBr) (Fig. 4d), where R-CH$_2$-IBr shows a ~1.50-fold higher barrier than that of IBr (1.01 and 1.51 eV, respectively). This implies that R-CH$_2$-IBr has enhanced anti-hydrolysis, which will enable enhanced I$_2$ utilization and prolonged battery lifespan. Self-discharge tests further substantiate this point (Supplementary Fig. S36). After resting for 12 h at a fully charged state, Zn||I$_2$ batteries using BrAce electrolytes exhibit a lower self-discharge rate, maintaining a CE of 77.44%. Conversely, batteries using ZnBr$_2$ electrolytes undergo a CE decay to 42.6%, accompanied by the complete disappearance of the discharge plateau of I$^0$/I$^+$ redox. This observation indicates the high susceptibility of IBr to hydrolysis. In contrast, the majority of the discharge plateau of

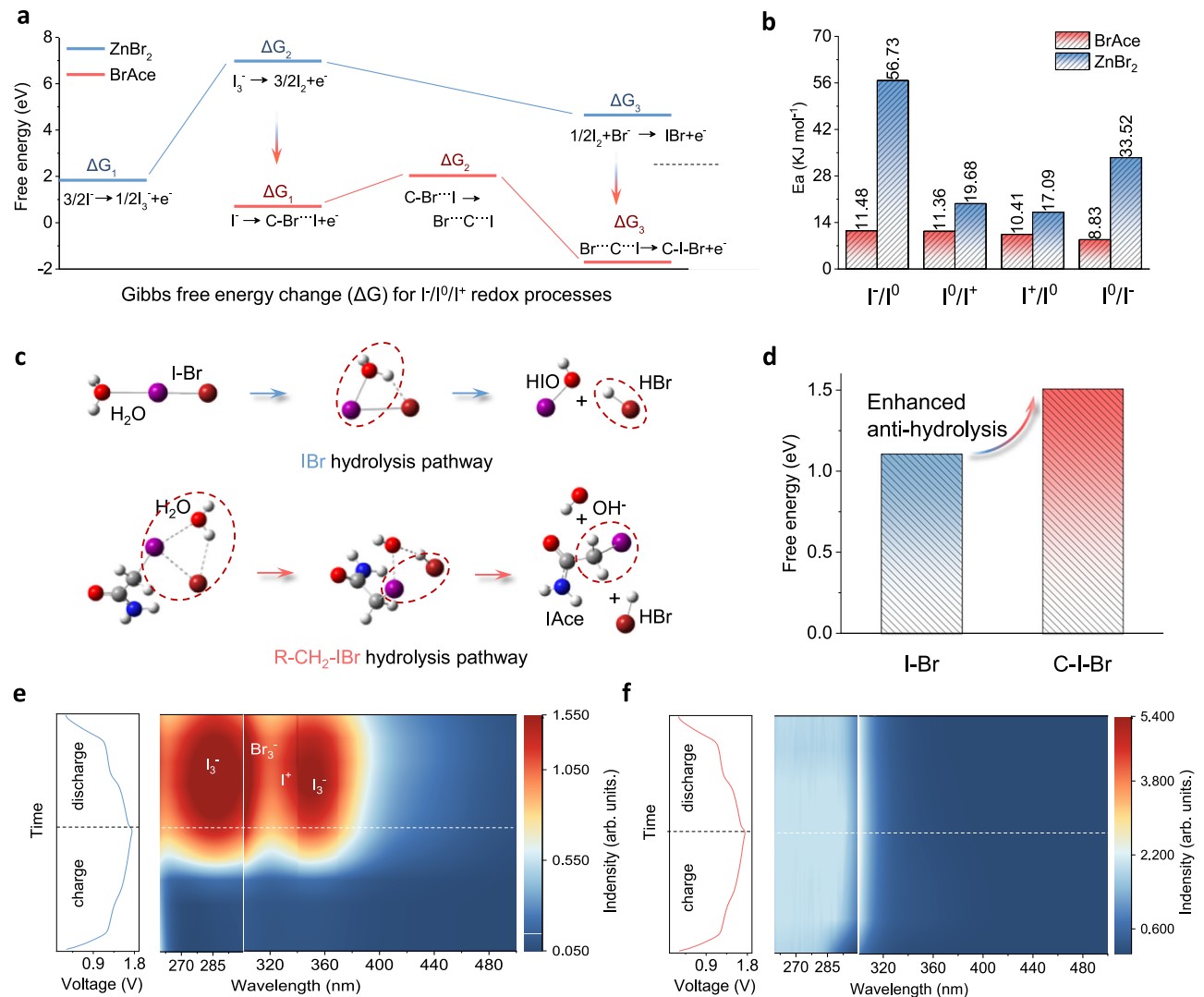

**Fig. 4 | Energy barriers for formation and hydrolysis of charged products.**
**a** Gibbs free energy change ($\Delta G$) for $I^-/I^0/I^+$ redox processes regulated by IBr and R-CH$_2$-IBr. **b** Comparison of $E_a$ values for $I^-/I^0$ and $I^0/I^+$ conversion regulated by IBr and R-CH$_2$-IBr. **c** Hydrolysis pathways of $I^+$ species (charged products). Red spheres: O atoms; Blue spheres: N atoms; Brown spheres: Br atoms; Purple spheres: I atoms; Gray spheres: C atoms; White spheres: H atoms. Circle: denotes emphasis. **d** The corresponding energy barriers. **e, f** In situ UV-vis spectra of electrolytes with ZnBr$_2$ and BrAce during charging/discharging of Zn‖I$_2$ batteries (cycling current: 0.2 A g$^{-1}$).

$I^0/I^+$ redox is preserved in BrAce electrolytes, verifying the stronger anti-hydrolysis capability of R-CH$_2$-IBr.

In situ ultraviolet-visible (UV-Vis) spectral characterization is further conducted with a cuvette cell (Supplementary Fig. S37). As revealed by *in situ* UV-vis spectra, $I_3^-$ (*ca.* 280 and 350 nm) and $Br_3^-$ (*ca.* 295 nm) appear in ZnBr$_2$ electrolytes, during discharging of batteries (Fig. 4e, Supplementary Fig. S38)[35,44,45]. In contrast, no $I_3^-$ or $Br_3^-$ signals can be detected in BrAce electrolytes (Fig. 4f, Supplementary Fig. S39). These results also correspond to the color change of electrolytes cycled (inset in Fig. 2f). The absence of $I_3^-$ or $Br_3^-$ in BrAce electrolytes could result from enhanced anti-hydrolysis of R-CH$_2$-IBr, which is beneficial for shuttle-free $I^-/I^0/I^+$ redox.

When certain methyl groups were introduced on the amide side as additives (e.g., 2-bromo-N-methylacetamide, 2-bromo-N, N-dimethylacetamide) in place of BrAce, although $I^0/I^+$ redox could be triggered, the capacity and I$_2$ utilization of the Zn‖I$_2$ batteries at 2 A g$^{-1}$ are lower (~ 310 and 260 mA h g$^{-1}$, 73.8% and 61.6%, respectively) (Supplementary Fig. S40). In addition, when using N-bromoacetamide, an additive with N-Br bonds, the capacity retention of the cell is only 37.5% after 500 cycles at 2 A g$^{-1}$ (Supplementary Fig. S41), which further confirms the distinct efficacy of the C-Br bond regulation and the advantages of the BrAce.

## Zn‖I$_2$ batteries with high I$_2$ loadings

Zn‖I$_2$ batteries involving $I^-/I^0/I^+$ redox are usually evaluated with relatively low I$_2$ loading (Supplementary Fig. S42). We examine the long-term cycling performance of Zn‖I$_2$ batteries with conventional I$_2$ loading (~ 3.0 mg cm$^{-2}$). During 2000 cycles at a small specific current of 2 A g$^{-1}$, Zn‖I$_2$ batteries in BrAce electrolytes show stable cycling performance, while those in ZnBr$_2$ electrolytes experience rapid capacity decay (Supplementary Figs. S43 and S44). At specific currents (10, 20, and 40 A g$^{-1}$), Zn‖I$_2$ batteries in BrAce electrolytes show higher capacities and higher I$_2$ utilization than those in ZnBr$_2$ electrolytes (Supplementary Figs. S45 and S46). Specifically, Zn‖I$_2$ batteries in BrAce electrolytes can be operated over 13 000 cycles with 82.9% retention and 66.0% I$_2$ utilization at 10 A g$^{-1}$, over 8000 cycles with 95.2% retention and 61.3% I$_2$ utilization at 20 A g$^{-1}$, over 16000 cycles with 95.6% retention and 55.2% I$_2$ utilization at 40 A g$^{-1}$, respectively. In contrast, I$_2$ utilizations of Zn‖I$_2$ batteries in ZnBr$_2$ electrolytes almost halve, only 35.8%, 35.2%, and 28.2% at 10, 20, and 40 A g$^{-1}$, respectively.

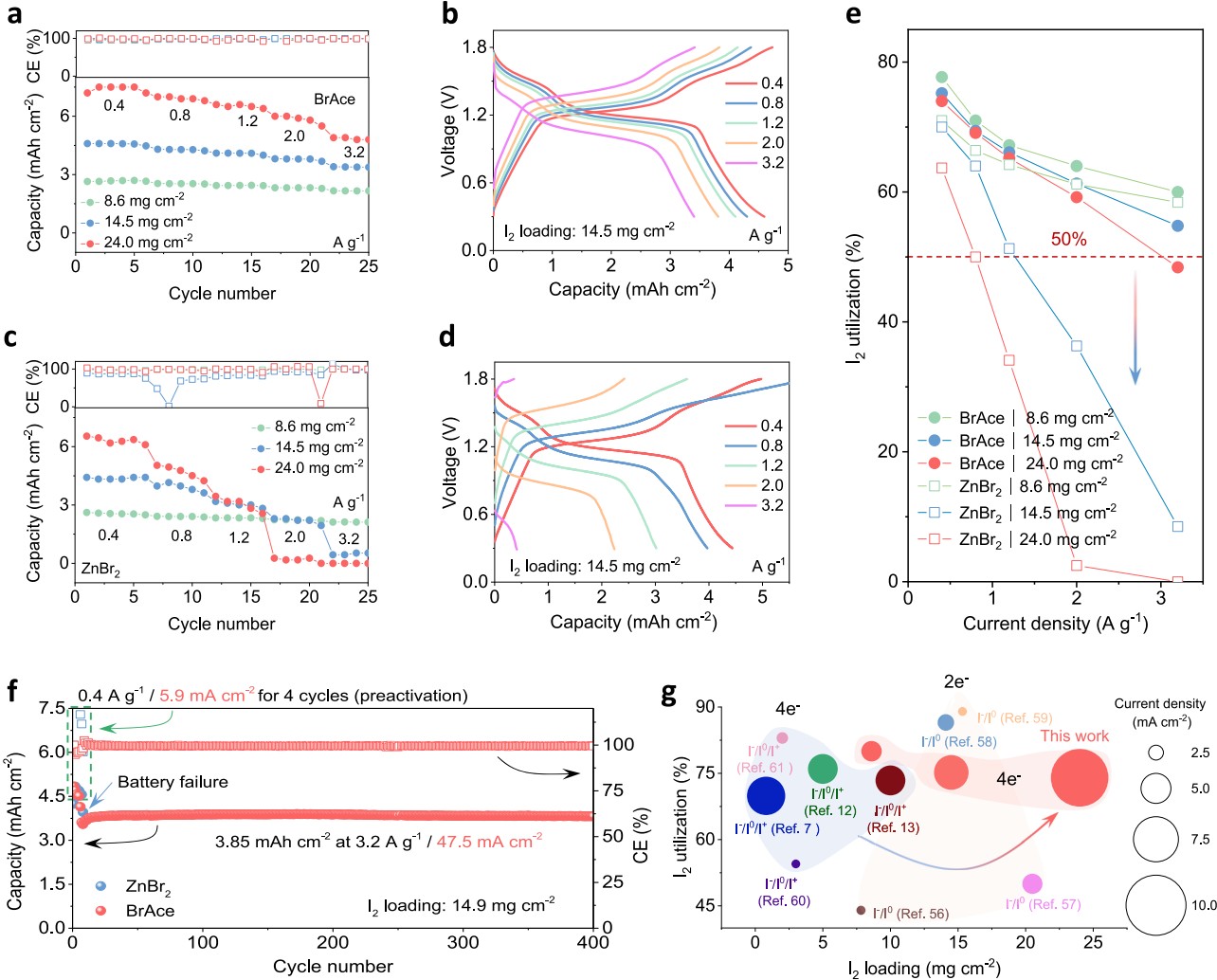

**Fig. 5 | Electrochemical performance of Zn∥I2 batteries with high I2 loadings.** **a**, **c** Rate capability of Zn∥I2 batteries in BrAce electrolytes and ZnBr2 electrolytes with different I2 loadings (8.6, 14.5, and 24.0 mg cm⁻²). **b**, **d** GCD curves of Zn∥I2 batteries in BrAce electrolytes and ZnBr2 electrolytes with I2 loadings of 14.5 mg cm⁻². **e** I2 utilizations (based on four-electron conversion capacity) of Zn∥I2 batteries with high I2 loadings in BrAce and ZnBr2 electrolytes at different specific currents.

**f** Long-term cyclability of Zn∥I2 batteries in BrAce and ZnBr2 electrolytes at 3.2 A g⁻¹/47.5 mA cm⁻² under high I2 loading (~14.9 mg cm⁻²). **g** Performance comparison of Zn∥I2 batteries with high I2 loading using BrAce electrolyte (red shading) with other reported Zn∥I2 batteries involving I⁻/I⁰ (two-electron transfer, yellow shading) and I⁻/I⁰/I⁺ (four-electron transfer, blue shading) redox reactions.

Moreover, for Zn∥I2 batteries in BrAce electrolytes, dual plateaus corresponding to I⁻/I⁰ and I⁰/I⁺ redox are well-kept at different cycles. Compared with other Zn∥I2 batteries involving I⁻/I⁰/I⁺ redox reported elsewhere, Zn∥I2 batteries in BrAce electrolytes display a certain degree of merits, in terms of rate capability and long-term cycling performance (Supplementary Fig. S47).

The effect of BrAce on Zn negative electrodes is scrutinized. In ZnBr2 electrolytes, Zn negative electrode of Zn∥I2 batteries (I2 loading: ~3 mg cm⁻²) cycled experiences severe pitting corrosion (Supplementary Fig. S48); while that in BrAce electrolytes remains intact. Compared with ZSO and ZnBr2 electrolytes, BrAce electrolytes effectively lower corrosion, as substantiated by Tafel curves at 1 mV s⁻¹ (Supplementary Fig. S49). CV curves of Zn∥Cu cells, long-term Zn plating/stripping performance of Zn∥Cu and Zn∥Zn cells suggest enhanced kinetics and prolonged cycle life by BrAce (Supplementary Figs. S50 and S51). Zn negative electrodes cycled in BrAce electrolytes show smooth surfaces, as revealed by SEM and LSCM images (Supplementary Figs. S52 and S53). Also, surficial passivation of the Zn negative electrode is alleviated by BrAce as revealed by XRD patterns (Supplementary Fig. S54). These confirm that the

efficacy of BrAce can be extended to Zn negative electrode modulation.

Due to severe shuttle and/or hydrolysis of I⁺ species, high-loading implementation of I⁰/I⁺ redox usually faces low utilization and short cycle life. To explore the loading limit, we first evaluated the rate capabilities of Zn∥I2 batteries with high I2 loadings (8.6, 14.5, and 24.0 mg cm⁻²). SEM image reveals electrode thicknesses of 374 and 493 μm, respectively, corresponding to I2 loadings of 8.6 and 14.5 mg cm⁻² (Supplementary Fig. S55). In BrAce electrolytes, Zn∥I2 batteries with different I2 loadings show favorable rate capabilities (Fig. 5a, Supplementary Fig. 56a). Specifically, for the I2 loading of 14.5 mg cm⁻², specific capacities of 4.6, 4.3, 4.1, 3.8, and 3.5 mA h cm⁻² are obtained at 0.4, 0.8, 1.2, 2.0, and 3.2 A g⁻¹ (areal current densities of 5.8, 11.6, 17.4, 29.0, and 46.4 mA cm⁻²), respectively. The corresponding GCD curves exhibit distinct plateaus for I⁰/I⁺ redox even at high rates (≥ A g⁻¹) (Fig. 5b). In contrast, in ZnBr2 electrolytes, despite similar rate capability for the I2 loading of 8.6 mg cm⁻² (Supplementary Fig. 57), severe capacity decay appears with higher I2 loadings (14.5 and 24.0 mg cm⁻²), especially at high rates (≥ A g⁻¹) (Fig. 5c, Supplementary Fig. 56b). In the corresponding GCD curves,

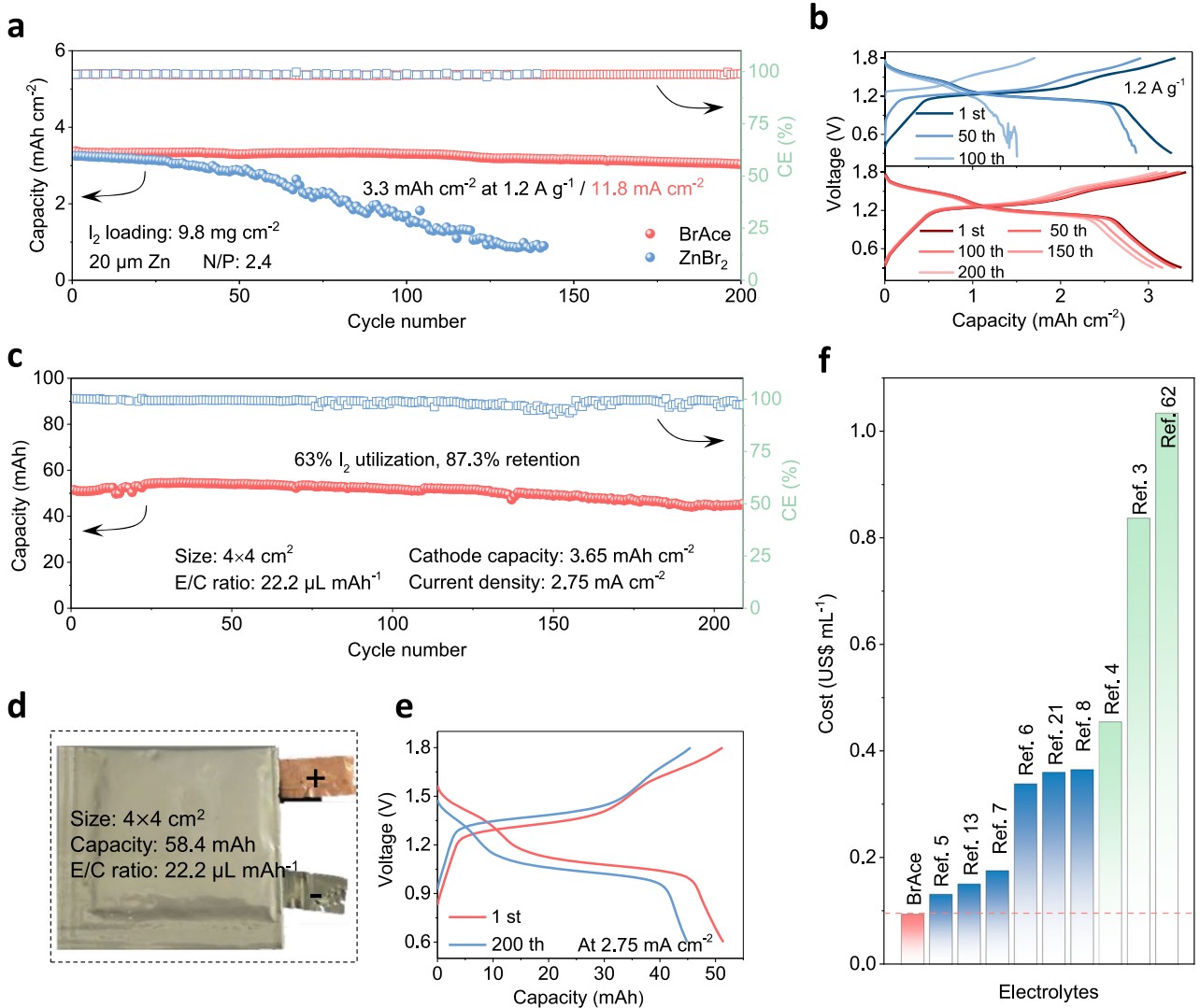

**Fig. 6 | Electrochemical performance of Zn∥I2 batteries under practical conditions. a** Long-term cyclability of Zn∥I$_2$ batteries in BrAce and ZnBr$_2$ electrolytes at 1.2 A g$^{-1}$/11.8 mA cm$^{-2}$ under high I$_2$ loading (~9.8 mg cm$^{-2}$) and low N/P ratio (~2.4). **b** The corresponding GCD curves at different cycles. **c** Long-term cyclability of Zn∥I$_2$ pouch cell in BrAce electrolytes at 2.75 mA cm$^{-2}$ with low E/C ratio (~22.2 μL mAh$^{-1}$). **d** The corresponding photograph of the tested pouch cells. **e** The corresponding GCD curves at different cycles. **f** Cost comparison of typical electrolytes for I$^-$/I$^0$/I$^+$ conversion, including BrAce electrolytes, highly concentrated salts, and hybrid electrolytes.

plateaus for I$^0$/I$^+$ redox almost disappear and severe polarization appear at high rates (≥ 1. A g$^{-1}$) (Fig. 5d, Supplementary Fig. 58), which is caused by sluggish I$^-$/I$^0$/I$^+$ redox kinetics. Moreover, Zn∥I$_2$ batteries with high I$_2$ loadings display enhanced I$_2$ utilization in BrAce electrolytes relative to ZnBr$_2$ electrolytes (Fig. 5e). Specifically, at the I$_2$ loading of 24 mg cm$^{-2}$, the I$_2$ utilization of the Zn∥I$_2$ battery is 65.2% vs. 34.2% at 1.2 A g$^{-1}$, and 59.2% vs. 2.5% at 2 A g$^{-1}$. These results verify the kinetics advantage of carbon-halogen bond substitution in compatibility with high I$_2$ loadings.

We further evaluated the long-term cyclability of Zn∥I$_2$ batteries with an I$_2$ loading of 14.9 mg cm$^{-2}$. Over 400 cycles at 3.2 A g$^{-1}$/ 47.5 mA cm$^{-2}$, an area-specific capacity of 3.85 mA h cm$^{-2}$ is obtained with BrAce electrolytes, corresponding to a 99.8% retention and 61.5% I$_2$ utilization (Fig. 5f). The corresponding differential charge/discharge profiles also confirm well-kept I$^-$/I$^0$/I$^+$ redox reversibility (Supplementary Fig. S59). In contrast, Zn∥I$_2$ batteries in ZnBr$_2$ electrolyte exhibit rapid capacity fade (~80% loss during only 7 cycles), followed by short-circuiting, highlighting the efficacy of BrAce. Compared with other Zn∥I$_2$ batteries (involving I$^-$/I$^0$ and I$^-$/I$^0$/I$^+$ redox) reported elsewhere, the Zn∥I$_2$ batteries in BrAce electrolytes exhibit advantages in

realizing high I$_2$ utilizations under high I$_2$ loadings at high rates (Fig. 5g, Supplementary Table S4)[7,12,13,56–61].

## Zn∥I2 batteries under practical conditions

To scrutinize the practicality of BrAce electrolytes, Zn∥I$_2$ batteries under practical conditions are further evaluated. At a low negative/ positive electrode capacity (N/P) ratio, Zn∥I$_2$ batteries in BrAce electrolytes (I$_2$ loading: 9.8 mg cm$^{-2}$; N/P ratio: 2.4) show stability over 200 cycles at 1.2 A g$^{-1}$/11.8 mA cm$^{-2}$ with an area specific capacity of 3.3 mA h cm$^{-2}$, corresponding to 90.4% retention and 98.5% CE (Fig. 6a). In contrast, Zn∥I$_2$ batteries in ZnBr$_2$ electrolytes appear rapid capacity decay after initial 25 cycles. The corresponding GCD curves at different cycles (1st, 50th, 100th, 150th, and 200th) show well-kept dual plateaus in BrAce electrolytes (Fig. 6b, bottom); while in ZnBr$_2$ electrolytes, the discharge curves appear evident fluctuation since the 50th cycle (Fig. 6b, top), which could be caused by accelerated pitting corrosion of thin Zn negative electrode (~20 μm).

Pouch cells with different high I$_2$ loadings are further assembled in BrAce electrolytes. Figure 6d shows the digital photograph of a pouch cell with a theoretical capacity of 58.4 mAh, size of 4 × cm$^2$, and

electrolyte/positive electrode (cathode) (E/C) ratio of 22.2 μL mAh$^{-1}$. After 210 cycles at 2.75 mA cm$^{-2}$, this pouch cell delivers an actual capacity of 54.5 mAh, corresponding to 63% I$_2$ utilization and 87.3% retention (Fig. 6c). The corresponding GCD curves at different cycles show well-kept dual plateaus (Fig. 6e). In addition, long-term cycling performance of pouch cell with a theoretical capacity of 35 mAh is evaluated. This pouch cell also shows stability over 312 cycles at 3.125 mA cm$^{-2}$, corresponding to 66.6% I$_2$ utilization and 86% retention (Supplementary Fig. S60). These results substantiate the practicality of pouch cells with BrAce electrolytes. Based on the commercial cost of components, the cost of BrAce electrolyte is as low as 0.1 US$ mL$^{-1}$, which is lower than those of the electrolytes (e.g., highly concentrated salts, hybrid electrolytes) for I$^-$/I$^0$/I$^+$ redox (Fig. 6f, Supplementary Table S5)[3–8,13,21,62]. The positive electrode fabrication employs ambient-temperature wet processing with iodine powder and activated carbon (AC), circumventing expensive catalysts or carbon materials (e.g., carbon nanotubes). This approach delivers economic advantages through simplified processing and scalable production pathways. Compared to organic electrolytes, the aqueous ZnSO$_4$–BrAce system exhibits higher ionic conductivity, lower viscosity, non-flammability, reduced toxicity, and minimal ecological footprint. The cost advantage and environmental impact advantage of BrAce electrolyte will pave the way for its application.

## Discussion

We discovered a carbon-halogen bond substitution pathway, enabling high-utilization cascaded iodine redox in noncorrosive and dilute electrolytes for sustainable four-electron Zn‖I$_2$ batteries. This pathway is realized with a low-cost, noncorrosive organohalide additive (BrAce), featuring a reversible Br-C···I$^{(0)}$ and C-I$^{(+)}$-Br bond substitution during I$^0$/I$^+$ conversion, as revealed by comprehensive spectral characterization. The criterion for additives with a universal formation of R-CH$_2$-Br (R=electron-withdrawing groups) is identified as the Hammett parameter ($\sigma$) of R, where an equal $\sigma$ value to that of -Br seems to be optimal. By modulating the axial $\delta$ hole activity of interhalogen bonding (I$^{(\delta+)}$-Br), this pathway synchronously lowers the barrier for I$^0$/I$^+$ redox and enhances the anti-hydrolysis of I$^+$ species. Compared with conventional interhalogen bonding (I-Br) pathway, this pathway enables sustainable operation of four-electron Zn‖I$_2$ batteries with high I$_2$ loading: (1) high I$_2$ utilizations (55%~80%) at high rates (5.8~46.4 mA cm$^{-2}$), (2) long lifespan (>400 cycles) with areal-specific capacity (~3.85 mA h cm$^{-2}$) and 99.5% retention at a high current density (47.5 mA cm$^{-2}$). This work not only unlocks a universal pathway for high-utilization cascaded iodine redox but also pioneers an additive design paradigm for scalable, high-energy, sustainable aqueous batteries beyond iodine.

## Methods

### Chemicals

All reagents and starting materials were used as received without any further purification. Zn foil (99.9%, 20 and 100 μm) and Cu foil (99.9%, 12 μm) were purchased from Shenzhen Kejing Star Technology. Zinc bromide (ZnBr$_2$, 99.9%), zinc sulfate heptahydrate (ZnSO$_4$·7H$_2$O, 99%), 2-(Bromomethyl)Tetrahydrofuran (BrTHF, 95%), and acetamide (Ace, 99%) were purchased from Aladdin. 2-bromoacetamide (BrAce, 97%), 3-bromopropionamide (3-BrPace, 97%), 2-chloroacetamide (2-ClAce, 97%), 2, 2-difluoroacetamide (DFAce, 98%), iodoacetamide (IAce, 98%), 2-bromo-N-methylacetamide (97%), 2-bromo-N,N-dimethylacetamide (97%), N-bromoacetamide (97%), Bromoacetonitrile (BrAN, 97%), Bromopropanone (BrPK, 95%), and IBr (98%) were purchased from Macklin. Deuterated chloroform (CDCl$_3$) and deuterated water (D$_2$O) were purchased from Sigma Aldrich.

### Electrolytes preparation

3 M (mol L$^{-1}$) ZnSO$_4$ solution (ZSO) was prepared by adding 3 M ZnSO$_4$ to 1 mL of deionized water and stirring for 2 h at 25 ± 1 °C.

Then, 0.35 M ZnBr$_2$ and 0.7 M BrAce were added to the as-prepared ZSO solutions, respectively, and stirred for 2~4 h at 25 ± 1 °C to obtain ZnBr$_2$ electrolytes and BrAce electrolytes. Similarly, other additives (e.g., Ace, BrAN, and BrTHF) were also configured into electrolytes at 0.7 M concentration for comparative investigations. All preparations were conducted in an ambient atmosphere.

### Battery assembly

The I@AC composite material was made by sufficiently grinding the activated carbon (AC, 99.99%, Canrd) with iodine powders (99.99%, Aladdin) at a mass ratio of 1:1. The I@AC, Super P (99.9%, Canrd), and sodium carboxymethyl cellulose binder (CMC aqueous solution, 99.99%, Anaiji) at a mass ratio of 8:1:1 were mixed by grinding to a slurry and drop-cast onto the carbon cloths current collector (Suzhou Sineros Technology), and then dried (8 h in an oven at 60 °C) to obtain the I@AC electrode (single-side coated). The dried electrode was then punched into disks with a diameter of 12 mm using a precision disc cutting machine (Shenzhen Kejing Star Technology, MSK-T10). The average areal I$_2$ loading of the I$_2$@AC composite was ~2.5 – 3.0 mg cm$^{-2}$ in the electrode. For the iodine-free positive electrode, the proportion of iodine was completely replaced by AC, while all other conditions remained unchanged. Metal foils (Zn, Cu) were also punched into 12 mm-diameter circular discs using the same precision disc cutting machine. Unless the temperature and environment are explicitly specified, all preparations were conducted in an ambient atmosphere at 25 ± 1 °C.

All cells were assembled in the configuration of CR2032-type coin cells. The cell structure was composed of two working electrodes and a glass fiber separator (Whatman GF/D, 16 mm in diameter, 675 μm in thickness); the remaining components included a stainless-steel spacer (1.0 mm thick) and a spring-loaded plunger (stainless steel, with a constant force of 1.5 kN). The Zn‖I$_2$ batteries were assembled using the as-obtained ZSO solution with different additives as the electrolytes, the I@AC electrodes as the positive electrodes, the metallic Zn foils (100 μm) as the negative electrodes, glass fibers as the separators. For the high depth of discharge (DOD) tests, the Zn foils with a thickness of 20 μm were used as the negative electrodes. The average areal I$_2$ loading of the I$_2$@AC composite was ~8.6 – 24.0 mg cm$^{-2}$ in the electrode.

For the fabrication of single-layer pouch cells, the same slurry mentioned above was coated onto 4 cm × 4 cm carbon cloths, with the mass loading of I$_2$ controlled at 13 mg·cm$^{-2}$. The assembled electrodes (Zn foil: 100 μm, 4 cm × 4 cm) were placed into an aluminum-laminated bag and vacuum-sealed, with one electrolyte injection port reserved for electrolyte filling. Approximately 1.3 mL of electrolyte was injected through this port, followed by a second vacuum-sealing step. The cells were then rested for 6 h to ensure sufficient wetting of all components by the electrolyte. During the battery cycling test, an external pressure of 1.2 MPa was applied to the pouch cells using a fixture. The pouch cells were tested at a cutoff voltage of 0.6–1.8 V and a current density of 2.75 mA·cm$^{-2}$.

### Material characterizations

The ionic conductivities of electrolytes were measured by a conductivity meter (DDS-308A). XRD analysis was used to determine the phase structures of the negative electrode samples with a Rigaku instrument (Cu Kα, λ = 1.5418 Å). The measurements were conducted at a scan rate of 5–10 °·min$^{-1}$ and a step size of 0.02 °. Infrared spectra of samples were recorded on an FT-IR spectrometer (Nicolet iS50R, Thermo Scientific). Raman spectra (HORIBA, LabRAM HR800) of the powder samples were measured at 25 ± 1 °C with laser excitation at 532 nm. The scanning electron microscope (SEM, Hitachi S-3400 N, beam voltage: 5–15 kV) with X-ray energy dispersive spectroscopy (EDS) was used to observe the morphology and elemental composition of the samples. Roughness of the surface of the resultant sample using

laser scanning confocal microscopy (LSCM). The $^1H$ and $^{13}C$ NMR spectra were recorded by a Bruker AVANCE III 500 MHz Superconducting Fourier using $CDCl_3$ as the solvent and trimethylsilane (TMS) as the standard internal substance. For ex situ $^1H$ and $^{13}C$ NMR spectra tests, the positive electrodes with different states of charge (SOC) were soaked in 0.6 mL $CDCl_3$ for 30 min and filtered to obtain a clear solution, then transferred to the NMR tube for analysis. The UV-vis spectra were collected by a UV-visible near-infrared spectrophotometer (Shimadzu). The chemical states of the electrodes after tests were determined by XPS (Thermo Fisher-Escalab 250Xi), and binding energies were calibrated using carbon (284.8 eV for C 1$s$). The mass spectrum was collected by Thermo Fisher Scientific Q Exactive Focus to qualitative determination of the formation of hypervalent iodine products.

### Preparation and characterization of the ex situ samples
For ex situ XPS, NMR, and mass spectrometry measurements, the cells were disassembled in an ambient atmosphere to investigate the surface chemical properties of the electrodes after cycling. After 3 cycles at a specific current of $0.5\,A\,g^{-1}$, the cycled iodine electrodes (at different states of charge) were harvested and rinsed three times with deionized water to remove residual electrolyte salts. Subsequently, the samples were dried at 60 °C for 1 h. The dried electrode samples were then transferred to the characterization instruments using a dedicated airtight transfer vessel. All characterization tests were conducted at $25 \pm 1$ °C. For the characterization of the negative electrode, the cycling was performed for 50 cycles under the same specific current, with all other procedures remaining unchanged.

### Preparation and characterization of the in situ samples
For in situ Raman, ATR-IR, and UV-Vis spectroscopy measurements, the iodine electrodes were first discharged to 0.3 V at a specific current of $0.5\,A\,g^{-1}$ or $0.2\,A\,g^{-1}$, followed by monitoring the variations of the electrodes over 1 full cycle (at different states of charge). All characterization tests were conducted at $25 \pm 1$ °C in an air atmosphere.

### Theoretical simulation calculation
**Free energy calculation by DFT**. Density functional theory (DFT) chemical description for the geometry optimization, the electronic structure calculations, including energies, and frequencies of all the stationary points (reactants, transition states, products) were carried out by the Gaussian09 program package with M062X exchange-correlation functional and 6-311 + G** basis set. Single energy calculations were performed by the aug-cc-pVTZ basis set.

**Equilibrium potential calculation**. The change in the Gibbs free energy ($\Delta G$) can be calculated by using the electron-transfer numbers ($n$) and the difference in the electrochemical potential ($\Delta U$).

$$\Delta G = -ne\Delta U$$

where $n$ is the electron transfer number, $e$ is the elemental charge, and $\Delta U$ is the electrochemical potential difference.

**Calculation of hydrolysis energy barriers**. Hydrolysis energy barriers (HEB) for I-Br bonds or C-I-Br bonds were calculated using the M062X/def2-TZVP level. The calculation formula is as follows:

$$A + H_2O \rightarrow B\cdot + C + D$$

$$HEB = G_B + G_C + G_D - G_A - G(H_2O)$$

where $G_A$ is the Gibbs free energy value of I-Br or C-I-Br, and $G_B$, $G_C$, and $G_D$ correspond to the Gibbs free energy values of the hydrolysis products of I-Br or C-I-Br in water attack, respectively.

### Electrochemical measurements
For the assembly of Zn||Zn symmetric cells, two identical Zn foils were sandwiched between a piece of glass fiber separator and then assembled into a coin cell under an ambient atmosphere. The Zn||Cu half-cells were fabricated similarly to the symmetric cells, with a cutoff voltage of 0.5 V. In the three-electrode system, Zn foil, Pt plate, and Ag/AgCl were used as the working, counter, and reference electrodes, respectively. These electrodes were utilized to perform LSV, Tafel and CV measurements on a Gamry electrochemical workstation. For LSV measurements: the corresponding scan rate was $1\,mV\cdot s^{-1}$, and the cutoff voltage was -0.6–2.5 V. For CV measurements: the corresponding scan rates were in the range of $0.5$–$20\,mV\cdot s^{-1}$, and the cutoff voltage was 0.3–1.75 V. GCD tests were carried out on a Neware battery test system (model CT-4008-10V50mA-164, from Shenzhen, China) within an electrochemical window of 0.3 – 1.75 V. EIS measurements were recorded by a Gamry electrochemical workstation, with a frequency range spanning from 0.01 Hz to 100 kHz. The measurements were performed in potentiostatic mode, with a sinusoidal amplitude of 10 mV, and 10 data points collected per frequency decade. Before each measurement, the cell was stabilized at the open-circuit potential (OCP) for at least 1 h until the voltage change was less than 1 mV per minute.

All batteries in this study were assembled and tested in an ambient atmosphere at $25 \pm 1$ °C. The capacity retention in the cycling test is calculated as the ratio of the discharge capacity of the last cycle to the discharge capacity of the first cycle. The calculation formula for self-discharge is the ratio of the discharge capacity after resting to the charge capacity of the charging cycle prior to resting. All data were obtained from at least three replicate measurements.

## Data availability
All data supporting the findings of this study are included in the Article and its Supplementary Information. Source data are provided with this paper. The atomic coordinates of the optimized geometries, along with related information, can be found in the Supplementary Data. Source data are provided with this paper.

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

## Acknowledgements

This research was funded by National Natural Science Foundation of China (52371216, 22305037, U24A20569, 52271204), Science and Technology Foundation of Shenzhen (JCYJ20190808153609561), Guangdong Basic and Applied Basic Research Foundation (2024A1515011920, 2023A1515030173, 2021A1515110952), Guangdong Provincial Key Laboratory of Plant Resources Biorefinery (2021B1212010011), Open Research Fund of Songshan Lake Materials Laboratory (2021SLABFN04), and Research Start-up Fund of GDUT (263113425). The authors also would like to thank the Analysis and Test Center of Guangdong University of Technology for the NMR and FTIR measurements.

## Author contributions

Conceptualization: Z. S. and Y. T. Data curation: Z. S. and Y. T. Formal analysis: Z. S., G. L., J. Q., H. H. and Z. F. Funding acquisition: Y. T., Y. W. and C. L. Investigation: Z. S., G. L., J. Q., H. H., Z. F. and Y. T. Methodology: Z. S., Y. W. and Y. T. Project administration: Y. T. and C. L. Resources: Y. T. and C. L. Supervision: Y. T., Y. W., W. D., M. Y., Y. Z., Z. W., X. L., Q. Y., C. Z., and C. L. Validation: Z. S. and Y. T. Visualization: Z. S., Y. T. and C. L. Writing - original draft: Z. S. and Y. T. Writing - review and editing: Z. S., C. L., C. Z. and Y. T.

## Competing interests

The authors declare no competing interests.
