## [Transparent Peer Review file · Nature Communications]

Carbon-Halogen Bond Substitution Enables High-Utilization Four-Electron Iodine Redox in Noncorrosive Dilute Electrolytes

Corresponding Author: Professor Chunyi Zhi

Version 0:

Reviewer comments:

Reviewer #1

(Remarks to the Author)

This manuscript proposes the four electron Zn-I₂ battery chemistry based on carbon-halogen bond substitution mediated by an organohalide additive. The authors argue that reversible Br-C...I(0) interactions and subsequent C-I⁺-Br bond formation drive the I⁻/I⁺ redox sequence. While the electrochemical measurements are consistent with a four electron iodine transfer, the study provides no direct spectroscopic or structural evidence to distinguish true C-I⁺-Br covalent bonding from simple interhalogen (I-Br) coordination. Consequently, the interpretation of iodine transformation lacks sufficient mechanistic justification, so the present manuscript is unsuitable for Nature Communications. Here are some comments should be considered.

1. Based on the charge distribution of covalent bonds, the electron-withdrawing induced effect of amide groups will affect the charge distribution of I⁺, then affect the charge distribution of IBr bonds, resulting in the shift of the spectral peak of IBr. How to understand the phenomenon that Raman signal of C-I-Br is consistent with IBr peak signal of I⁺-Br- (Fig. 3a and Fig. 3b)? Meanwhile, the Raman peak of C-I-Br should be furtherly confirmed by standard samples.

2. What is the electrochemical performance of BrAce in the tested voltage window without iodine?

3. As long as the bromine in the additive can participate in the iodine conversion process, the NMR signal will change, which only shows that the additive is coordinated, but can not explain the formation of C-I(+)-Br bond. Please provide visual data such as XPS or mass spectrometry to prove the formation of C-I(+)-Br bond.

4. In the ex-situ ¹H NMR and ¹³C NMR, the NMR curves of D-1.4V and D-1.2V are almost consistent with the Pristine curve of BrAce, but obviously different from those of C-1.4V and C-1.2V (Line 190-193, Fig. 3d-3e.). However, the electrode products at 1.2-1.4V should be the same, so how to understand that the NMR results of the same product are quite different during charging and discharging? In addition, the current NMR results can not prove the reversible formation of -CH₂-Br...I₀ and I₀...(CH₂)...Br during discharge, so it is suggested to supplement the in-situ NMR characterization.

5. Line 214. If I₀ plays the role of electron donor in I₀...(CH₂)...Br, the charge density of methylene will increase, so the signals of methylene of both ¹H-NMR and ¹³C-NMR should shift to higher field, not the lower field (Figure. 3d C-1.2V, C-1.5V). Therefore, please consider the contribution of I₀ to the charge distribution again. The electrostatic potential of I₀...(CH₂)...Br should be further provided.

6. When different organic additives are introduced, the redox peaks of iodine are obviously different in Fig. S23. Why are the redox peaks of organic additives so close to those of zinc bromide in Fig. S13? This is unreasonable.

7. The article employs DFT calculations to determine hydrolysis energy barriers for the I⁰/I⁺ redox process. However, the calculations do not account for the influence of the electrolyte environment (e.g., ZnSO₄) on the structural stability of R-CH₂-IBr. While the energy described in Fig. 4d represents the hydrolysis energy barrier of IBr and C-I-Br, but not their free energy. On this diagram, the reader mistakenly thinks that the free energy of IBr is less than that of C-I-Br, which means that C-I-Br has worse stability. Please correct it and provide supplementary details addressing this aspect. Meanwhile, please provide relevant experiments to prove the improvement of hydrolysis stability.

8. The authors claim that the battery exhibits excellent conversion kinetics; however, the GCD profiles reveal that the capacity contribution from I⁺/I₀ under high iodine loading is small (as evidenced by the markedly abbreviated voltage plateau) in Fig. 5b, Fig. 5d and Fig. 6e. This diminished plateau suggests that the I₀→ I⁺ oxidation step may be kinetically limited.

9. In Line 223, the group R disappears in the product. Please standardize the writing of equation Step2. Many footnote and other formats in this paper are incorrect, please modify them.

Reviewer #2

(Remarks to the Author)

This work addresses critical challenges hindering practical aqueous zinc-iodine batteries, especially the hydrolysis/shuttle of high-valent I^+ species and sluggish kinetics in multi-electron iodine conversion. The authors present that a carbon-halogen bond substitution mechanism enabling stable four-electron (I^-/I^+) conversion chemistry. This mechanism operates effectively in non-corrosive, dilute electrolytes. This approach may circumvent the inherent limitations of previously required high-concentration halide salt systems. The reported electrolyte achieves excellent performance with near-commercial cathode utilization even under high iodine loadings. As such, I recommend this manuscript be published after addressing the following comments:

1. The authors only show the role of 0.3-1 M BrAce in participating in the four-electron iodine conversion reaction; does the performance change at higher concentrations of BrAce?
2. Since the redox potentials of I^0/I^+ and Br^-/Br^0 are relatively close, could more evidence be provided to show that no Br conversion reactions are involved?
3. The authors mention in Fig. 2a-c that BrAce can promote iodine conversion attributed to the presence of a carbon-bromine bond. However, BrPK and BrAN with similar molecule structures to BrAce seem less effective. Does the nitrogen atom/group play a role in iodine conversion?
4. It has been confirmed that the salting-out effect of SO_4^{2-} can inhibit the shuttle of polyiodides (e.g., I_3^- , I_5^-). In this work, the authors used baseline electrolytes with 3 M zinc sulfate. Does the concentration of zinc sulfate influence the stability of I^0/I^+ ?
5. The authors need to explain the reason why the differences in iodine utilization between the full cells assembled with the addition of BrAce and $ZnBr_2$ are so significant in Fig. 2c, while the differences are not evident in Figs. 5 and 6.
6. The authors mention that the carbon-halogen bond coordination confers fast kinetics to the iodine conversion reactions. However, based on the b-values and capacitive contributions in Figs. S14 and S15, the differences between BrAce and $ZnBr_2$ are not significant. Please provide further explanations and clarifications.
7. There are some typos in the supplementary information (e.g., Fig. S38). Please double-check and correct.

Reviewer #3

(Remarks to the Author)

Comment on NCOMMS-25-35778: (Carbon-Halogen Bond Substitution Enables High-Utilization Cascaded Iodine Redox in Noncorrosive Dilute Electrolytes)

The manuscript presents an innovative approach to enhancing iodine utilization in $Zn||I_2$ batteries via carbon-halogen bond substitution using a C-Br functionalized organic additive (BrAce). This study addresses critical challenges associated with iodine redox reactions in dilute, noncorrosive electrolytes. The authors have provided substantial experimental and theoretical evidence to support their claims. However, a few areas require additional clarification and improvement before acceptance for publication.

1. The manuscript compellingly introduces a carbon-halogen bond substitution mechanism. Nevertheless, direct experimental evidence, such as mass spectrometry, would considerably reinforce the hypothesized intermediate CH_2-IBr .
2. In Fig. 2d, I_2 utilization does not increase with the Hammett parameter σ . BrAce ($\sigma = 0.28$) exhibits the highest utilization, while additives with higher σ values, such as BrAN and BrPK, perform less effectively. This suggests there may be an optimal σ range. A brief discussion on why stronger electron-withdrawing groups might hinder activation would enhance the mechanistic interpretation.
3. Can the author further clarify the NMR spectra peak between 1 and 2 ppm in Fig. 3d? For the 1H -NMR spectrum of BrAce, it should display two distinct signals at 2.37 and 3.97 ppm, corresponding to two different types of protons in the molecule, respectively.
4. While the manuscript demonstrates strong practical applicability, further discussion of scalability, manufacturing feasibility, and potential environmental impacts would significantly enhance the study's comprehensiveness.
5. Improving the clarity of figures by explicitly labeling schematic pathways, as shown in Fig. 3f, would help readers follow the proposed mechanism.
6. Please correct the inconsistent abbreviation of bromoacetonitrile (BrCN) throughout the figures in the manuscript.

Reviewer #4

(Remarks to the Author)

Version 1:

Reviewer comments:

Reviewer #1

(Remarks to the Author)

I appreciate the authors' efforts to elucidate the proposed C–I bond formation. However, the collected ^1H and ^{13}C NMR spectra at various states of charge and other supplemented experiments do not provide convincing evidence for the claimed C–I(+)-Br covalent bond, which is the central mechanism of this study. If such a covalent bond were indeed formed, the C–I(+)-Br moiety should behave analogously to an alkyl iodine species in organic chemistry. In that case, it would be expected to exhibit characteristic spectroscopic signatures and chemical reactivity distinct from those of simple IBr. The absence of these features casts doubt on the proposed structural assignment.

Furthermore, if the “electron-withdrawing amide group of BrAce diminishes electron density in adjacent atoms,” then the adjacent I+ center should be less polarized. A less polarized I+ would be expected to exhibit a lower valence state compared to IBr, which contradicts both the XPS data and the ^{13}C NMR results. In addition, such a less polarized I+ in the C–I(+)-Br structure should yield a lower discharge voltage than conventional IBr, which is not observed.

The self-discharge tests also raise concerns: the I+/I2 plateau is nearly absent in the charge–discharge curves, suggesting that BrAce electrolytes fail to stabilize I+ effectively. Moreover, under high iodine loading, the capacity contribution from the I+/I0 redox process is minimal, indicating that the I0 → I+ oxidation step may be kinetically hindered.

The comparison of voltage polarization reported in Figure 2e is not fair.

Given these issues, the mechanistic interpretation of iodine transformations remains insufficiently justified. I regret that I cannot provide a positive assessment of this work again.

Reviewer #2

(Remarks to the Author)

The authors have addressed my comments successfully.

Reviewer #3

(Remarks to the Author)

Comment on NCOMMS-25-35778A: (Carbon-Halogen Bond Substitution Enables High-Utilization Cascaded Iodine Redox in Noncorrosive Dilute Electrolytes)

The authors have made appropriate revisions according to the suggestions. Therefore, I suggest acceptance for publication.

1. Line 114/532: change “Hametter” to “Hammett”.
2. Lines 230/235: ATR-IR units should be cm^{-1} , not nm; please check for consistency.
3. Replace “electron-drawing” with “electron-withdrawing” in the text and Fig. 1 as they are different in meaning.
4. Supplementary figures: change “figure 50” to “Supplementary Fig. S50” and apply the same style to later figures.

Reviewer #4

(Remarks to the Author)

Version 2:

Reviewer comments:

Reviewer #1

(Remarks to the Author)

I don't have other comments. The manuscript can be published.

Response Letter

Dear Reviewers,

Thank you very much for your message pertaining to our submission. We have studied the comments and suggestions. In our point-by-point responses, the Reviewers' comments are presented in black, our responses are provided in **Blue**. The modifications implemented during the revision are indicated in both the revised manuscript and the Supporting Information, highlighted in **Yellow**. The following is a summary of our revisions and responses to referees.

Reviewer #1:

This manuscript proposes the four electron Zn-I₂ battery chemistry based on carbon-halogen bond substitution mediated by an organohalide additive. The authors argue that reversible Br-C···I⁽⁰⁾ interactions and subsequent C-I⁺-Br bond formation drive the I⁻/I⁰/I⁺ redox sequence. While the electrochemical measurements are consistent with a four electron iodine transfer, the study provides no direct spectroscopic or structural evidence to distinguish true C-I⁺-Br covalent bonding from simple interhalogen (I-Br) coordination. Consequently, the interpretation of iodine transformation lacks sufficient mechanistic justification, so the present manuscript is unsuitable for Nature Communications. Here are some comments should be considered.

Reply:

We sincerely appreciate the reviewer's professional evaluation of our work. We wish to emphasize the fundamental novelty of this study: it represents the first demonstration of a C-Br bond-containing additive enabling the I⁰/I⁺ redox chemistry within a noncorrosive low-concentration salt electrolyte, ultimately achieving four-electron transfer in Zn batteries. Indeed, the carbon-halogen substitution coordination pathway lacks direct precedent in the literature, presenting significant challenges in mechanistic validation. Hence, we acknowledge the reviewer's concerns as critically important areas requiring diligent examination. To address these points, we have

provided comprehensive supplementary explanations, experimental validations, and spectroscopic evidence within our revised manuscript and this response.

1. Based on the charge distribution of covalent bonds, the electron-withdrawing induced effect of amide groups will affect the charge distribution of I^+ , then affect the charge distribution of I-Br bonds, resulting in the shift of the spectral peak of I-Br. How to understand the phenomenon that Raman signal of C-I-Br is consistent with I-Br peak signal of I^+-Br^- (Fig. 3a and Fig. 3b)? Meanwhile, the Raman peak of C-I-Br should be furtherly confirmed by standard samples.

Reply:

Thanks for your insightful question and valuable suggestion. We have provided supplementary data and mechanistic interpretations regarding the difference between Raman signals of I-Br and C-I-Br species, through theoretical and experimental analysis.

When the electron-withdrawing amide group is directly involved in the vibration, it may change the polarity of the bond and thus the position of the C-I-Br peak. Hence, we agree the reviewer's opinion. Actually, compared with conventional I-Br, the signal of C-I-Br does appear a red shift (**Fig. R1**), which is consistent with the electron-withdrawing inductive effect. We are sorry about having not made this point clear. Meanwhile, it is worth noting that the *in-situ* Raman spectra under identical experimental conditions reveal substantially lowered peak intensity for C-I-Br compared to I-Br (**Fig. 3a,b**). Usually, the intensity of Raman scattering is fundamentally governed by the magnitude of polarizability change during molecular vibration (Biomacromolecules 2020, 21, 3485–3497). In our work, electron-withdrawing amide group of BrAce can diminish electron density in adjacent atoms or functional groups, effectively “fixing” the electron cloud. This fixation can reduce the dynamic polarizability during vibrational modes, thereby weakening Raman scattering intensity for C-I-Br bonds. Moreover, the C-I-Br intermediate demonstrates exceptional stability, without detectable polyiodide or polybromide signals. This contrasts markedly with conventional I-Br bond pathway that generates substantial side products,

confirming distinct differences in chemical properties between them.

This relevant data and explanation have been added to revised manuscript (Page 7) and supporting information (Page 8). Please see the highlighted parts.

Figure R1. Raman spectra of I-Br and C-I-Br intermediates.

Regarding the reviewer's query about C-I-Br reference standards, we politely clarify that there are no direct standard samples available so far for comparison. Since the C-I-Br is an intermediate during the I⁰/I⁺ redox via a novel carbon-halogen bond substitution pathway. Alternatively, we have performed a series of spectroscopic characterization (Raman spectra, XPS spectra, ¹H/¹³C NMR, etc.) to confirm the presence of C-I-Br structure. Please see Reply to Comments 3.

2. What is the electrochemical performance of BrAce in the tested voltage window without iodine?

Reply:

Thanks for the thoughtful question. We have conducted the test using cathode without iodine in the same electrolyte and voltage window. The cathode was prepared by the processes: AC, Super P, and sodium carboxymethylcellulose binder (CMC aqueous solution) in a mass ratio of 8:1:1 was mixed to form a slurry, and then coated on carbon cloth (collector) and dried (8 h in an oven at 60 degrees Celsius). The testing parameters were kept in line with those in the main article (current density: 1 A g⁻¹; carbon loading: 2.5 mg cm⁻²; voltage window: 0.3 -1.75 V). As shown in **Fig. R2**, the Galvanostatic charge/discharge (GCD) curves display a typical capacitive behavior, in which no bromine redox conversion can be identified, suggesting no bromine redox

interference in four-electron iodine conversion. Meanwhile, the pure carbon positive electrode maintains good stability after 1000 cycles, which is consistent with the capacitive characteristics.

This relevant data and explanation have been added to revised manuscript (Page 5) and supporting information (Page 8). Please see the highlighted parts.

Fig. R2 (a) GCD curves of the iodine-free battery assembled with BrAce electrolyte at a current density of 1 A g⁻¹; (b) the corresponding long-term cycling performance.

3. As long as the bromine in the additive can participate in the iodine conversion process, the NMR signal will change, which only shows that the additive is coordinated, but can not explain the formation of C-I⁽⁺⁾-Br bond. Please provide visual data such as XPS or mass spectrometry to prove the formation of C-I⁽⁺⁾-Br bond.

Reply:

Thank you for your thoughtful suggestions. To prove the formation of C-I⁽⁺⁾-Br bond, we have provided the XPS and mass spectrometry and made comparative analysis.

Fig. R3 shows the high-resolution XPS spectra of Br 3d, I 3d, and C 1s for the C-I-Br samples at different charge/discharge potentials. **Fig. R4** shows the high-resolution XPS spectra of Br 3d, I 3d, and C 1s for the I-Br samples at different charge/discharge potentials. As shown in **Fig R3**, the XPS spectra of both Br 3d and I 3d of samples at high potentials (C-1.75 V, D-1.6 V) appear the signals associated with I⁺ species at 74.0 and 631.5 eV, respectively, which correspond to the characteristic of I⁰/I⁺ redox confirmed by CV and GCD curves in **Fig. 2b-c**. Both the binding energies for I⁺ species

in Br 3d and I 3d XPS spectra are evidently different from those for the counterparts (70.3 eV and 631.8 eV) in samples via conventional I-Br pathway (**Fig. R5**). Specifically, the I^+ species in Br 3d of C-I-Br shows a 3.7-eV higher binding energy than that of I-Br. Such a large gap manifests that, in addition to bonding between Br and I, the electron density around the Br also remarkably reduces, which could be caused by inductive effect of electron-withdrawing group (amide). This is consistent with the result revealed by *in-situ* Raman spectra in **Fig. 3a, b**, i.e., the C-I-Br pathway remarkably lowers the polarizability of charged product.

To confirm the presence of C-(I-Br) moiety, we further conducted the liquid chromatography-mass spectrometry (LC-MS) characterization of samples via C-I-Br pathway. As shown in **Fig R6**, the strong signal appears at m/z 264.8, corresponding to the weight of $NH_2-C(O)-CH_2-(IBr)$, thereby confirming the presence of C-(I-Br) moiety.

To further clarify the binding structure of C-I-Br, we further compared the 1H NMR of C-I-Br samples with commercial IAce (C-I bond) and BrAce (C-Br bond). As shown in **Fig R7**, the featured signal of alpha-hydrogen attached to IBr in C-I-Br typically appear between those of IAce and BrAce. Notably, this signal is much closer to that of IAce but far away from that BrAce, which verifies similar chemical properties of C-I-Br samples to IAce. Therefore, logically, the alpha-carbon is more likely to bond with I rather than Br.

Such results collectively confirm the formation of C- $I^{(+)}$ -Br bond. We have added the corresponding explanation in the revised manuscript (Page 8, 10) and supporting information (Page 9-11). Please check. Limited by harsh characterization conditions, we have done our best to address your concern, and hope to satisfy you. Many thanks!

Fig. R3 *Ex-situ* XPS spectra of cathode during GCD in BrAce electrolyte: (a) Br 3d spectra, (b) I 3d spectra, and (c) C 1s spectra.

Fig. R4 *Ex-situ* XPS spectra of cathode during GCD in ZnBr₂ electrolyte: (a) Br 3d spectra and (b) I 3d spectra.

Fig. R5 *Ex-situ* XPS spectra of the cathode charged to 1.75 V in ZnBr₂ and BrAce electrolytes: (a) Br 3d spectra and (b) I 3d spectra.

Fig. R6 Mass spectrometry of C-I-Br sample (C-1.75 V).

Fig. R7 ^1H NMR spectra of IAce, C-I-Br sample (C-1.75 V) and IAce.

4. In the ex-situ ^1H NMR and ^{13}C NMR, the NMR curves of D-1.4 V and D-1.2 V are almost consistent with the Pristine curve of BrAce, but obviously different from those of C-1.4 V and C-1.2 V (Line 190-193, Fig. 3d-3e.). However, the electrode products at 1.2-1.4 V should be the same, so how to understand that the NMR results of the same product are quite different during charging and discharging? In addition, the current NMR results can not prove the reversible formation of $-\text{CH}_2\text{-Br}\cdots\text{I}^0$ and $\text{I}^0\cdots(\text{CH}_2)\cdots\text{Br}$ during discharge, so it is suggested to supplement the in-situ NMR characterization.

Reply:

Thanks for your insightful questions and thoughtful suggestions. As illustrated in **Fig. 3d-e**, the initial step of the I^-/I^0 conversion during charging involves adsorption of iodide ions around C-Br bonds (Step 1: $2\text{I}^- - 2\text{e}^- + \text{R-CH}_2\text{-Br} \rightarrow 2\text{R-CH}_2\text{-Br}\cdots\text{I}^0$, where $\text{R} = \text{NH}_2\text{-(C=O)-}$). This mechanistic feature fundamentally suppresses polyiodide formation in BrAce-modified $\text{Zn}||\text{I}_2$ batteries. Subsequent oxidation at the I^-/I^0 redox stage (C-1.75 V) triggers substitution coordination, leading to halogen exchange (Step 3: $2\text{I}^0\cdots(\text{CH}_2)\cdots\text{Br} - 2\text{e}^- \rightarrow 2\text{R-CH}_2\text{-I}^+\text{-Br}$). This transition manifests in ^1H NMR spectra through a pronounced chemical shift displacement of the $-\text{CH}_2\text{Br}$ proton signal (originally at 3.65 ppm) to the characteristic $-\text{CH}_2\text{I}$ resonance (3.62 ppm), as corroborated in **Fig. S21**.

During discharge, the I^+/I^0 reduction proceeds via $2R-CH_2-I^+-Br + 2e^- \rightarrow 2R-CH_2Br + I_2$, followed by $I_2 + 2e^- \rightarrow 2I^-$ at the I^0/I^- stage. The diminished involvement of C-Br bonds in the discharge process accounts for spectral discrepancies between charged (C-1.4 V, C-1.2 V) and discharged (D-1.4 V, D-1.2 V) states. Crucially, the preservation of the $-CH_2-I^+-Br$ signal at the high-voltage plateau (C-1.75 V/D-1.6 V) and its full reversion to the original $-CH_2Br$ signal upon deep discharge (D-1.4 V/D-1.2 V) verifies electrochemical reversibility of BrAce. Complementary evidence arises from: (i) *in-situ* Raman spectra (**Fig. 3a-b**) demonstrating reversible I-Br bond formation without Br_3^-/I_3^- byproducts; (ii) FT-IR spectra of $-CH_2Br \rightleftharpoons -CH_2IBr$ interconversion (**Fig. 3c**). Such data collectively validate our revised reaction scheme (Step 4: $2R-CH_2-I^{(+)}-Br + 2e^- \rightarrow 2R-CH_2Br + I_2$; Step 5: $I_2 + 2e^- \rightarrow 2I^-$).

As for *in-situ* NMR characterization, we politely clarify that current technical constraints do preclude its implementation. While widely deployed in Li-ion batteries, aqueous Zn|| I_2 batteries present intractable challenges: 1) Magnetic field distortion: Zn dendrites formed during plating/stripping generate heterogeneous magnetic gradients, compromising spectral fidelity; 2) Signal interference: Electrolyte anions (SO_4^{2-} , Br^-) exhibit $^1H/^{17}O$ resonances overlapping critical analytes (e.g., SO_4^{2-} -associated H_2O at δ 4.5 ppm obscures $-CH_2Br$ protons); 3) Low sensitivity & side reactions: The natural abundance of ^{13}C (1.1%) and dilute C-Br concentration (0.7 M) necessitate >1-hour acquisition per state. Prolonged testing at minimal currents will exacerbate electrolyte decomposition.

Based on the above considerations, we are currently unable to overcome this technical bottleneck. However, our NMR spectroscopic characterization has been repeated more than twice to confirm the accuracy of the analysis. Limited by harsh characterization conditions, we have done our best to address your concern, and hope to satisfy you. Many thanks! We have added the corresponding explanation in the revised manuscript (Page 10-11). Please check.

5. Line 214. If I^0 plays the role of electron donor in $I^0 \cdots (CH_2) \cdots Br$, the charge density of methylene will increase, so the signals of methylene of both 1H -NMR and ^{13}C -NMR

should shift to higher field, not the lower field (Figure. 3d C-1.2 V, C-1.5 V). Therefore, please consider the contribution of I⁰ to the charge distribution again. The electrostatic potential of I⁰⋯(CH₂)⋯Br should be further provided.

Reply:

Thanks for your valuable question and suggestion. We concur with your opinion on the iodine redox mechanism. As depicted in **Fig. 3d-e**, the electrochemical evolution during charging from 1.2 V to 1.4 V (stage ①→②) initiates the I⁻/I⁰ conversion, where iodine species (I⁰) function as Lewis acids that initially coordinate with the bromine atom of the -CH₂-Br moiety. Given the high polarizability of I⁰, this Br⋯I coordination could facilitate electron withdrawal from the Br-C bond through n→σ* antibonding orbital interactions. Consequently, the methylene group experiences deshielding, accounting for the downfield shifts in both ¹H and ¹³C NMR spectra.

Subsequently, during the voltage elevated to 1.5 V (stage ②→③), the -CH₂-Br⋯I⁰ undergoes isomerization to form I⁰⋯CH₂⋯Br structure. In this rearranged state, I⁰ transitions into an electron donor, cooperatively enhancing electron density at the methylene carbon alongside bromine. This electronic redistribution elevates shielding effects, thereby inducing upfield shifts of the methylene proton and carbon resonances in NMR spectra. We acknowledge that this mechanistic nuance was not explicitly clarified in our original manuscript.

Following your suggestion, we have conducted supporting electrostatic potential simulations to further corroborate this dynamic behavior. As shown in **Fig. R8**, when the CH₂Br⋯I⁰ intermediate forms, iodine accumulates electron density while depleting electron density around the methylene group, rationalizing the initial downfield NMR shifts. Following isomerization to the I⁰⋯(CH₂)⋯Br intermediate, I⁰ donates electron density to the methylene carbon, increasing its electron cloud density and inducing the upfield shift. These computational results align with the experimental NMR spectra.

We have added the corresponding explanation in the revised manuscript (Page 9-10) and supporting information (Page 11). Please check.

Fig. R8 Electrostatic potential in different reaction states. (a) BrAce, (b) CH₂Br...I⁰, and (c) the I⁰... (CH₂) ...Br.

6. When different organic additives are introduced, the redox peaks of iodine are obviously different in Fig. S23. Why are the redox peaks of organic additives so close to those of zinc bromide in Fig. S13? This is unreasonable.

Reply:

Thanks for your thoughtful questions.

Regarding Question 1: As shown in **Fig. S23**, when 2-bromo-N-methylacetamide and 2-bromo-N, N-dimethylacetamide are used as additives, significant polarization differences can be observed between the two redox couples. It suggests that the choice of organic additive substantially impacts the kinetics of the iodine redox reactions. This effect arises because the electron-withdrawing capabilities of amide groups differ among the organic additives (-CONHCH₃ > -CON(CH₃)₂). These differences alter the nucleophilicity of the bromine atom in the CH₂-Br moiety, consequently influencing its ability to bond with electrochemically generated I⁺ species.

Regarding Question 2: **Fig. S13** presents a comparison of CV curves at various scan rates for batteries employing the two different electrolytes, aiming to illustrate their kinetic differences. Since the iodine conversion mechanism remains a four-electron redox process (I⁻/I⁰/I⁺) in both electrolytes containing BrAce and ZnBr₂, the redox peak profiles of CV curves are similar. However, it is worth noting that distinct differences in the response currents and the electrochemical polarization of the redox couples are evident between the CV curves with the two electrolytes. These differences become particularly pronounced at higher scan rates (10-20 mV s⁻¹). This result clearly indicates that BrAce, as an additive, facilitates superior kinetics for activating the four-

electron iodine conversion compared to conventional ZnBr₂.

7. The article employs DFT calculations to determine hydrolysis energy barriers for the I⁰/I⁺ redox process. However, the calculations do not account for the influence of the electrolyte environment (e.g., ZnSO₄) on the structural stability of R-CH₂-IBr. While the energy described in Fig. 4d represents the hydrolysis energy barrier of IBr and C-I-Br, but not their free energy. On this diagram, the reader mistakenly thinks that the free energy of IBr is less than that of C-I-Br, which means that C-I-Br has worse stability. Please correct it and provide supplementary details addressing this aspect. Meanwhile, please provide relevant experiments to prove the improvement of hydrolysis stability.

Reply:

Thanks for your insightful questions and suggestions.

Firstly, to clarify the influence of SO₄²⁻ concentration on the stability of R-CH₂-IBr, we have tested Zn||I₂ batteries using 1 M ZnSO₄ + 0.7 M BrAce and 2 M ZnSO₄ + 0.7 M BrAce as the electrolytes, respectively. As shown in **Fig. R9**, these batteries both exhibited no significant capacity decay over 1000 cycles at 2 A g⁻¹, with specific capacities of 340~350 mAh g⁻¹ following an initial activation. This performance is notably higher than that achieved with several other C-Br bond additives mentioned in the manuscript. Both iodine utilization and cycling stability were significantly superior to the 3 M ZnSO₄ + ZnBr₂ system (**Fig. S27**) and close to that with 3 M ZnSO₄ + 0.7 M BrAce. These results confirm that I⁰/I⁺ redox process is primarily governed by the modulation effect of carbon-halogen bond substitution coordination rather than SO₄²⁻ concentration. Self-discharge performance tests further highlighted substantial differences in hydrolysis behavior between C-I-Br and I-Br species under identical ZnSO₄ electrolyte conditions (explained below). Consequently, DFT calculations determining the hydrolysis energy barrier for the I⁰/I⁺ redox reaction did not incorporate the influence of ZnSO₄.

Secondly, we politely emphasize that the energy values presented in **Fig. 4d** represent the hydrolysis energy barriers for I-Br and C-I-Br species, not the Gibbs free energy barriers for intermediates during I⁰/I⁺ conversion. Based on the hydrolysis

reaction pathway depicted in **Fig. 4c**, the hydrolysis energy barrier for C-I-Br was calculated as 1.51 eV. This value was derived by subtracting the free energy of the initial reactants from the free energy of the hydrolysis products of C-Br-I. In contrast, the barrier for I-Br hydrolysis was determined to be 1.01 eV. The higher barrier for C-I-Br implies its superior resistance to hydrolysis, consistent with interpretations in related literature (Sci. Bull. (2024) 69, 1674-1685). Relevant computational details and data have been added to the revised manuscript (Page 12) and supporting information (Page 16) for your review.

Finally, we supplemented the study with self-discharge tests, combined with characterizations in our manuscript, to demonstrate the enhanced anti-hydrolysis stability of C-I-Br. After resting for 12 hours at a fully charged state, Zn||I₂ batteries employing the BrAce electrolyte exhibited a lower self-discharge rate, maintaining a Coulombic efficiency (CE) of 77.44%. Conversely, batteries using ZnBr₂ electrolytes experienced significant CE decay to 42.6%, accompanied by the complete disappearance of the discharge plateau of I⁰/I⁺ redox. This observation indicates the high susceptibility of I-Br to hydrolysis. In contrast, the majority of the discharge plateau of I⁰/I⁺ redox was preserved in the BrAce electrolytes, verifying the stronger anti-hydrolysis capability of C-I-Br. Furthermore, *in-situ* UV-vis spectra (**Fig. 4e-f**) revealed strong signals characteristic of I₃⁻ (peaks around 280 nm and 350 nm) and Br₃⁻ (peak around 295 nm) in the ZnBr₂ electrolyte during battery discharge. Strikingly, neither I₃⁻ nor Br₃⁻ signals were detectable in the BrAce system. The absence of these polyhalides in the BrAce system is likely attributable to the enhanced anti-hydrolysis properties of R-CH₂-IBr, which enables a shuttle-free I⁻/I⁰/I⁺ redox process. This finding aligns with the observed color change in the cycled electrolytes (inset, **Fig. 2f**): the BrAce electrolyte showed no significant discoloration, while the ZnBr₂ electrolyte exhibited distinct yellowing.

Collectively, these results provide sufficient evidence for the significantly improved hydrolysis stability of C-I-Br relative to conventional I-Br. We have added the corresponding explanation in the revised manuscript (Page 12) and supporting information (Page 16). Please check.

Fig. R9 Cycling performance of Zn||I₂ batteries in (a) 1 M ZSO and (b) 2 M ZSO electrolytes at 2 A g⁻¹.

Fig. R10 Self-discharge using (a) ZnBr₂ and (b) BrAce electrolytes after standing for 12 h in fully charged state.

8. The authors claim that the battery exhibits excellent conversion kinetics; however, the GCD profiles reveal that the capacity contribution from I⁺/I⁰ under high iodine loading is small (as evidenced by the markedly abbreviated voltage plateau) in Fig. 5b, Fig. 5d and Fig. 6e. This diminished plateau suggests that the I⁰ → I⁺ oxidation step may be kinetically limited.

Reply:

Thank you for your valuable questions.

As shown in **Fig. 2e**, a significant difference in iodine utilization between Zn||I₂ batteries assembled with BrAce and ZnBr₂ electrolytes exists under low iodine loadings (2-3 mg cm⁻²). Specifically, at 1 A g⁻¹, the utilization reached 82.74% vs. 66.3%. This indicates that the reversibility of C-I-Br pathway substantially surpasses that of

conventional I-Br pathway at conventional iodine loadings. **Fig. 4a** reveals that the Gibbs free energy barrier for the traditional reaction pathway is considerably higher than that for the C-I-Br pathway. This is further supported by the disparity in activation energies tested across different charge/discharge states. When the current density was increased to 20 A g⁻¹ and 40 A g⁻¹, the gap in iodine utilization widened further to 55.4% vs. 33.6% and 52.9% vs. 26.3%, respectively. While high current densities limit the reversibility of iodine conversion, compared to other reported systems achieving four-electron iodine conversion, our work maintains stable high-voltage plateaus (I⁰/I⁺ redox) even under ultrahigh currents, demonstrating an exceptional conversion kinetics.

Under higher iodine loadings, iodine conversion utilization is usually constrained by aggravated side reactions and polyiodide shuttling. Consequently, few studies report four-electron iodine conversion at high iodine loadings. In our work, we retained A g⁻¹ as the unit for rate capability testing of high-loading Zn||I₂ batteries to facilitate direct comparison across different loadings. As shown in **Fig. 5b**, at an iodine loading of 14.5 mg cm⁻², the applied current densities of 1.2, 2, and 3.2 A g⁻¹ correspond to actual areal currents of 17.4, 29, and 46.4 mA cm⁻², respectively. While these high currents impede complete I⁰/I⁺ conversion, stable charge/discharge plateaus are still maintained, achieving utilizations of 66.1%, 61.3%, and 54.8%. In sharp contrast, the control sample (ZnBr₂ system, **Fig. 5d**) exhibits no discernible high-voltage plateau for I⁰/I⁺ conversion at the same current density (1.2 A g⁻¹) and fails to operate normally at higher currents. Its utilization dropped dramatically from 51.3% to 8.5%. The substantial difference in utilization between the two electrolytes at 24 mg cm⁻² iodine loading (**Fig. S41**) further corroborates this observation (1.2 A g⁻¹: 65.2% vs. 34.2%; 2 A g⁻¹: 59.2% vs. 2.5%). Therefore, under harsh high-loading conditions, the C-I-Br pathway demonstrates superior kinetics compared to conventional I-Br.

Scaling from coin cells to pouch cells usually can amplify the challenges associated with high iodine loadings. As shown in **Fig. 6e**, the I⁰/I⁺ conversion is somewhat constrained in the pouch cell with a 14 mg cm⁻² iodine loading. However, **Fig. S43** shows that a pouch cell with a 7.8 mg cm⁻² loading exhibits more complete I⁰/I⁺ redox plateaus. Moreover, the current fabrication process of aqueous zinc ion

batteries is still immature, and the problems of thick iodine electrodes being easy to fall off and crack are not effectively solved, which could compromise the I^0/I^+ redox kinetic under high loadings.

9. In Line 223, the group R disappears in the product. Please standardize the writing of equation Step 2. Many footnote and other formats in this paper are incorrect, please modify them.

Reply:

Thanks for your nice suggestions. We have standardized the writing of equation Step 2. In addition, we have modified the footnote and format errors, and double-checked the full paper to eliminate similar problems. Please check.

Reviewer #2:

This work addresses critical challenges hindering practical aqueous zinc-iodine batteries, especially the hydrolysis/shuttle of high-valent I^+ species and sluggish kinetics in multi-electron iodine conversion. The authors present that a carbon-halogen bond substitution mechanism enabling stable four-electron ($I^-/I^0/I^+$) conversion chemistry. This mechanism operates effectively in non-corrosive, dilute electrolytes. This approach may circumvent the inherent limitations of previously required high-concentration halide salt systems. The reported electrolyte achieves excellent performance with near-commercial cathode utilization even under high iodine loadings. As such, I recommend this manuscript be published after addressing the following comments:

Reply:

We greatly appreciate your positive and professional evaluation of our work. We have provided supplementary experimental data and corresponding explanations to address your concerns.

1. The authors only show the role of 0.3-1 M BrAce in participating in the four-electron iodine conversion reaction; does the performance change at higher concentrations of BrAce?

Reply:

During the preliminary screening of additive concentrations, we tried adding much more BrAce (e.g., 1.2 and 1.5 M) to the ZSO electrolytes, but it ultimately failed to dissolve at room temperature. Although the solubility of BrAce can be increased by heating and stirring, but the electrolyte returned to turbidity after 10 minutes of standing. Therefore, the soluble concentration limit of BrAce at room temperature was around 1 M. As shown in Supplementary **Fig. S11**, when the concentration of BrAce is increased to 0.7 M, the $Zn||I_2$ battery at $1 A g^{-1}$ delivers the highest capacity and the longest cycling stability. When the concentration is increased up to 1 M, the battery capacity appears decrease. It indicates that the nearly saturated BrAce can reduce the reversibility of iodine conversion to some extent, which could be associated with reduced mass transfer

kinetics by hydration of polar amide groups in the additives.

We have added the relevant explanation in the revised manuscript (Page 6). Please check.

2. Since the redox potentials of I^0/I^+ and Br^-/Br^0 are relatively close, could more evidence be provided to show that no Br conversion reactions are involved?

Reply:

Thanks for the valuable question. As shown in Supplementary Fig. R11, when the Zn||I₂ battery was further increased to 1.85 V, a new charging plateau appeared at 1.78 V, corresponding to the Br^-/Br^0 conversion; during discharging, the corresponding discharge plateau appeared, along with a significant decrease in the Coulombic efficiency. It suggests that excessively high charging voltage can cause electrolyte decomposition, leading to the irreversible fracture of C-Br bonds of additives. In contrast, when the charging voltage is kept below 1.75 V, the Br^-/Br^0 conversion can be eliminated, thereby ensuring the efficacy of BrAce during the $I^-/I^0/I^+$ redox. This point has been sufficiently verified by the GCD curves (Fig. 2b) and long-term cycling (Fig S28, S29).

We have added the corresponding GCD curves and explanation in the revised manuscript (Page 5) and supporting information (Page 7). Please check.

Fig. R11 GCD curve of a Zn||I₂ battery assembled with BrAce electrolyte charged to 1.85 V at a current density of 1 A g⁻¹.

Fig. S12 CV curves of Zn||I₂ batteries with BrAce electrolytes in a cut-off voltage of 0.3-2.0 V.

3. The authors mention in Fig. 2a-c that BrAce can promote iodine conversion attributed to the presence of a carbon-bromine bond. However, BrPK and BrAN with similar molecule structures to BrAce seem less effective. Does the nitrogen atom/group play a role in iodine conversion?

Reply:

We politely emphasize that, to some extent, the all the C-Br bond-containing additives with electron-withdrawing groups can activate and stabilize the four-electron iodine conversion (Fig. 2a-c). As shown in Supplementary Fig. S3-S4, the Zn||I₂ batteries assembled with BrAN and BrPK electrolytes at 2 A g⁻¹ exhibit decent cycling stability during 500 cycles. It confirms a good universality of the C-Br bond activation strategy towards four-electron iodine conversion.

Hammett parameter (σ) can be used to quantify the electron-withdrawing strength of the organic additives, i.e., a more positive σ value indicates greater electron-withdrawing ability (J. Phys. Chem. C 2024, 128, 29, 12178–12185). Compared to BrAce with amide group ($\sigma=0.28$), BrAN (nitrile, $\sigma=0.66$) and BrPK (ketone, $\sigma=0.51$) additives provide inferior I₂ utilization. In theory, the optimal additive should be able to bond with the activated I⁺ species rapidly, and stabilize the charged product. Excessive electron withdrawal can significantly diminish the nucleophilicity of the bromine atom within the C-Br bond, hindering its rapid reaction with newly formed I⁺

species. Consequently, while BrAN and BrPK additives provide cycling stability, the substantially higher σ values of their nitrile and ketone substituents reduce the bromine nucleophilicity of the C-Br bond. This insufficiently activates the iodine conversion reaction, resulting in diminished I₂ utilization. Additionally, Supplementary Fig. S1 also shows that the use of acetamide as an additive alone is unable to activate the conversion of I⁰/I⁺, thus ruling out a dominant role of nitrogen atoms/groups in iodine conversion.

We have added the corresponding explanation in the revised manuscript (Page 4-5). Please check.

4. It has been confirmed that the salting-out effect of SO₄²⁻ can inhibit the shuttle of polyiodides (e.g., I₃⁻, I₅⁻). In this work, the authors used baseline electrolytes with 3 M zinc sulfate. Does the concentration of zinc sulfate influence the stability of I⁰/I⁺?

Reply:

Thanks for your thoughtful question. We agree that the salting-out effect of SO₄²⁻ can inhibit the dissolution and shuttling of the iodine species, which could be beneficial to improve iodine utilization and cycling stability of batteries. To scrutinize the effect of SO₄²⁻ concentration, we have evaluated Zn||I₂ batteries using the electrolytes composed of 1 M ZSO+0.7 M BrAce and 2 M ZSO+0.7 M BrAce, respectively. As shown in Fig. R13, no evident capacity decay was observed over 1000 cycles at 2 A g⁻¹. The specific capacities of *ca.* 350 mA h g⁻¹ are achieved in both electrolytes, which are higher than several other C-Br bond-containing additives mentioned in this paper. Also, the iodine utilization was close to that of 3 M ZSO + 0.7 M BrAce electrolyte. It confirms that the iodine redox stability in our work mainly result from the modulation of BrAce (Fig. 4d), while the effect of ZSO concentration could be secondary.

We have added the corresponding explanation in the revised manuscript (Page 12) and supporting information (Page 16). Please check.

Fig. R9 Cycling performance of Zn||I₂ batteries in (a) 1M ZSO and (b) 2 M ZSO electrolytes at 2 A g⁻¹.

5. The authors need to explain the reason why the differences in iodine utilization between the full cells assembled with the addition of BrAce and ZnBr₂ are so significant in Fig. 2c, while the differences are not evident in Figs. 5 and 6.

Reply:

Thanks for your nice question. Actually, the iodine utilization is highly influenced by the current density and iodine loading, especially the latter. Under low iodine loadings (2-3 mg cm⁻²), **Fig. 2e** reveal a significant disparity (82.74% vs. 66.3% at 1 A g⁻¹) in iodine utilization for Zn||I₂ batteries assembled with BrAce and ZnBr₂ electrolytes. This performance gap is further widened substantially at higher rates of 20 and 40 A g⁻¹, where iodine utilization drops to 55.4% vs. 33.6% and 52.9% vs. 26.3%, respectively. The reduced iodine utilization at elevated current densities stems from restricted iodine conversion reversibility, aggravated side reactions, and iodine species shuttling.

For consistency in comparing rate capability across different loadings, current density (A g⁻¹) was maintained during high-loading Zn||I₂ battery testing. As shown in **Fig. 5b**, under a high iodine loading of 14.5 mg cm⁻², applied current densities of 1.2, 2, and 3.2 A g⁻¹ correspond to high areal current densities of 17.4, 29, and 46.4 mA cm⁻². While these large current densities impede complete I⁰/I⁺ redox, stable charge/discharge plateaus are still maintained, achieving iodine utilizations of 66.1%, 61.3%, and 54.8%, respectively. In sharp contrast, **Fig. 5d** demonstrates that the ZnBr₂

electrolyte exhibits no discernible high-plateau I^0/I^+ redox even at 1.2 A g^{-1} and fails to sustain normal operation at higher current densities, with utilization decrease from 51.3% to 8.5%. This profound difference is further corroborated in Supplementary **Fig. S52** at an extreme loading of 24 mg cm^{-2} , where utilizations are 65.2% vs. 34.2% at 1.2 A g^{-1} and 59.2% vs. 2.5% at 2 A g^{-1} .

These results collectively demonstrate the superior iodine utilization of the BrAce electrolyte compared to ZnBr_2 electrolyte, irrespective of the level of iodine loading or current density. We have added the corresponding explanation in the revised manuscript (Page 15). Please check.

6. The authors mention that the carbon-halogen bond coordination confers fast kinetics to the iodine conversion reactions. However, based on the b -values and capacitive contributions in Figs. S14 and S15, the differences between BrAce and ZnBr_2 are not significant. Please provide further explanations and clarifications.

Reply:

Thanks for your valuable question and suggestion. In our work, both the control and test electrolytes incorporated equimolar bromine-containing additives (0.35 M ZnBr_2 and 0.7 M BrAce) into baseline electrolytes (3 M ZnSO_4 , ZSO). Unlike conventional “water-in-salt” or hydrated eutectic electrolytes that stabilize iodine conversion reactions primarily by limiting water activity, this formulation achieves an effective balance between viscosity and ionic conductivity (**Fig. S9**), thereby retaining the inherently favorable kinetics characteristic of aqueous systems. Consequently, the calculated b -values and capacitive contributions based on CV curves (**Fig. S14-S15**) show minimal divergence between the electrolytes.

However, comparative analysis of the CV curves across different scan rates (1-20 mV s^{-1} , **Fig. S13**) reveals different kinetic behavior. The BrAce electrolyte exhibits significantly higher response currents at the redox peaks alongside reduced polarization of iodine redox couples. This polarization disparity becomes gradually more pronounced at faster scan rates. The larger voltage hysteresis observed in the ZnBr_2 electrolyte indicates inferior reaction reversibility and slower kinetics. Moreover, both

the Gibbs free energy calculated by DFT and activation energy obtained by Arrhenius equation confirm substantially lower iodine redox barrier by BrAce than ZnBr₂ (Fig. 4a-b).

These results collectively establish a fundamental difference in reaction kinetics governed by carbon-halogen coordination compared to traditional interhalogen bond coordination.

7. There are some typos in the supplementary information (e.g., Fig. S38). Please double-check and correct.

Reply:

Thank you for your suggestion. The typos in the supplementary information have been corrected. Please check.

Reviewer #3:

Comment on NCOMMS-25-35778: (Carbon-Halogen Bond Substitution Enables High-Utilization Cascaded Iodine Redox in Noncorrosive Dilute Electrolytes)

The manuscript presents an innovative approach to enhancing iodine utilization in Zn||I₂ batteries via carbon-halogen bond substitution using a C–Br functionalized organic additive (BrAce). This study addresses critical challenges associated with iodine redox reactions in dilute, noncorrosive electrolytes. The authors have provided substantial experimental and theoretical evidence to support their claims. However, a few areas require additional clarification and improvement before acceptance for publication.

Reply:

We greatly appreciate your positive and professional evaluation of our research. We have provided the following responses to address your concerns.

1. The manuscript compellingly introduces a carbon-halogen bond substitution mechanism. Nevertheless, direct experimental evidence, such as mass spectrometry, would considerably reinforce the hypothesized intermediate CH₂–IBr.

Reply:

Thanks for your insightful suggestion. To prove the formation of C–I⁽⁺⁾–Br bond, we have provided the XPS and mass spectrometry and made comparative analysis.

Fig. R3 shows the high-resolution XPS spectra of Br 3d, I 3d, and C 1s for the C–I–Br samples at different charge/discharge potentials. **Fig. R4** shows the high-resolution XPS spectra of Br 3d, I 3d, and C 1s for the I–Br samples at different charge/discharge potentials. As shown in **Fig R3**, the XPS spectra of both Br 3d and I 3d of samples at high potentials (C-1.75 V, D-1.6 V) appear the signals associated with I⁺ species at 74.0 and 631.5 eV, respectively, which correspond to the characteristic of I⁰/I⁺ redox confirmed by CV and GCD curves in **Fig. 2b-c**. Both the binding energies for I⁺ species in Br 3d and I 3d XPS spectra are evidently different from those for the counterparts (70.3 eV and 631.8 eV) in samples via conventional I–Br pathway (**Fig. R5**). Specifically, the I⁺ species in Br 3d of C–I–Br shows a 3.7-eV higher binding energy than that of I–Br. Such a large gap manifests that, in addition to bonding between Br

and I, the electron density around the Br also remarkably reduces, which could be caused by inductive effect of electron-withdrawing group (amide). This is consistent with the result revealed by *in situ* Raman spectra in Fig. 3a-b, i.e., the C-I-Br pathway remarkably lowers the polarizability of charged product.

To confirm the presence of C-(I-Br) moiety, we further conducted the liquid chromatography-mass spectrometry (LC-MS) characterization of samples via C-I-Br pathway. As shown in Fig R6, the strong signal appears at m/z 264.8, corresponding to the weight of $\text{NH}_2\text{-C(O)-CH}_2\text{-(I-Br)}$, thereby confirming the presence of C-(I-Br) moiety.

To further clarify the binding structure of C-I-Br, we further compared the ^1H NMR of C-I-Br samples with commercial IAce (C-I bond) and BrAce (C-Br bond). As shown in Fig R7, the featured signal of alpha-hydrogen attached to IBr in C-I-Br typically appear between those of IAce and BrAce. Notably, this signal is much closer to that of IAce but far away from that BrAce, which verifies similar chemical properties of C-I-Br samples to IAce. Therefore, logically, the alpha-carbon is more likely to bond with I rather than Br.

Such results collectively confirm the formation of C-I⁽⁺⁾-Br bond. We have added the corresponding explanation in the revised manuscript (Page 8, 10) and supporting information (Page 9-11). Please check.

Fig. R3 *Ex-situ* XPS spectra of cathode during GCD in BrAce electrolyte: (a) Br 3d

spectra, (b) I 3d spectra, and (C) C 1s spectra.

Fig. R4 *Ex-situ* XPS spectra of cathode during GCD in ZnBr_2 electrolyte: (a) Br 3d spectra and (b) I 3d spectra.

Fig. R5 *Ex-situ* XPS spectra of the cathode charged to 1.75 V in ZnBr_2 and BrAce electrolytes: (a) Br 3d spectra and (b) I 3d spectra.

Fig. R6 Mass spectrometry of C-I-Br sample (C-1.75 V).

Fig. R7 ^1H NMR spectra of IAce, C-I-Br (C-1.75 V), and IAce.

2. In Fig. 2d, I_2 utilization does not increase with the Hammett parameter σ . BrAce ($\sigma = 0.28$) exhibits the highest utilization, while additives with higher σ values, such as BrAN and BrPK, perform less effectively. This suggests there may be an optimal σ range. A brief discussion on why stronger electron-withdrawing groups might hinder activation would enhance the mechanistic interpretation.

Reply:

Thanks for your valuable suggestion. We agree that there is an optimal σ range for fast and stable I^0/I^+ redox. Hammett parameter (σ) can be used to quantify the electron-

withdrawing strength of the organic additives, i.e., a more positive σ value indicates greater electron-withdrawing ability (J. Phys. Chem. C 2024, 128, 29, 12178–12185). As shown in **Fig. 2d**, compared to BrAce with amide group ($\sigma=0.28$), BrAN (nitrile, $\sigma=0.66$) and BrPK (ketone, $\sigma=0.51$) additives provide inferior I_2 utilization. On the contrary, with 2-bromomethyltetrahydrofuran (furan, $\sigma=-0.11$) additive, the I^0/I^+ redox hardly occurs. Furthermore, when operated at a current density of 2 A g^{-1} , the battery shows rapid capacity fade to 75 mA h g^{-1} after 300 cycles. Although the GCD curves during cycling transiently show the I^0/I^+ plateau, the rapid capacity decay indicates poor product stability, contrasting sharply with additives bearing electron-withdrawing groups linked to the C-Br bond.

In theory, the optimal additive should be able to bond with the activated I^+ species rapidly, and stabilize the charged product. Excessive electron withdrawal can significantly diminish the nucleophilicity of the bromine atom within the C-Br bond, hindering its rapid reaction with newly formed I^+ species. Consequently, while BrAN and BrPK additives provide cycling stability, the substantially higher σ values of their nitrile and ketone substituents (compared to bromine itself, **Table S1**) reduce the bromine nucleophilicity of the C-Br bond. This insufficiently activates the iodine conversion reaction, resulting in diminished iodine utilization. Conversely, electron-donating groups do not prevent IBr formation but destabilize the charged products, thereby failing to effectively stabilize the I^0/I^+ redox couple. Interestingly, the amide group in BrAce shows a slightly higher σ value than that of Br (0.28 vs. 0.23). The slightly higher σ value of electron-withdrawing groups than that of bromine of C-Br bond could be more beneficial to achieve the trade-off for maximized iodine redox reversibility and superior I_2 utilization.

We have added the corresponding explanation in the revised manuscript and highlighted on Page 4-5. Please check.

3. Can the author further clarify the NMR spectra peak between 1 and 2 ppm in Fig. 3d? For the $^1\text{H-NMR}$ spectrum of BrAce, it should display two distinct signals at 2.37 and 3.97 ppm, corresponding to two different types of protons in the molecule, respectively.

Reply:

Thank you for your constructive suggestions. To identify this signal between 1 and 2 ppm of the ^1H NMR spectra, we have conducted a series of comparative NMR characterization. As shown in **Fig. R12a**, dissolution of BrAce and IAce in deuterated solvents (CDCl_3) reveals distinct proton signals for these halogenated acetamides. Both compounds exhibit two characteristic resonances at approximately 2.45 ppm and 3.3 ppm, attributable to the chemically inequivalent protons of the amide ($-\text{NH}_2$) group. This assignment is justified given the near-identical molecular structures of BrAce and IAce, differing solely in their $-\text{CH}_2\text{-X}$ substituents ($\text{X} = \text{Br}$ or I). The nonequivalence of amide protons arises from restricted rotation around the C-N bond, yielding two discrete signals. Notably, the methylene protons display divergent chemical shifts: the $-\text{CH}_2\text{I}$ moiety of IAce resonates at 3.56 ppm, while the counterpart $-\text{CH}_2\text{Br}$ of BrAce appears at 3.77 ppm. This differentiation aligns with **Fig. 3d**, wherein the $-\text{CH}_2\text{Br}$ signal undergoes a substantial displacement to 3.65 ppm during the I^0/I^+ redox transition at the charging stage.

To simulate operational environment of BrAce in aqueous batteries, we dissolved BrAce into deionized water for ^1H NMR test. As shown in **Fig. R12b-d**, the amide proton signals (2.45 ppm and 3.3 ppm) almost disappear due to the interference of hydrogen-bonding interaction between water and amide group. In contrast, the hydrophobic $-\text{CH}_2\text{Br}$ protons remained distinct at 3.8 ppm, demonstrating their relative insensitivity to aqueous solvation. Subsequently, we analyzed the cathode samples during GCD cycling. When charging beyond 1.2 V induced a new resonance at 1.18 ppm (**Fig. 3d**). These peaks experience analogous potential-dependent shifts to the $-\text{CH}_2\text{Br}$ signal at 3.65 ppm ($-\text{CH}_2\text{Br}$ proton) (**Fig. 3d**). We attribute this emergent signal to one proton of the $-\text{CH}_2\text{Br}$ group, where asymmetric electronic environments resulting from interaction (chemical adsorption or bonding) with iodine species decouple the methylene protons, yielding distinct chemical shifts. Such decoupling implies covalent or coordinative bonding between BrAce and iodine intermediates during charge transfer.

We have added the corresponding explanation in the revised manuscript (Page 8-9) and supporting information (Page 10). Please check.

Fig. R12 (a) ^1H NMR spectra of BrAce and IAce. (b)-(d) ^1H NMR spectra of BrAce and BrAce in deionized water.

4. While the manuscript demonstrates strong practical applicability, further discussion of scalability, manufacturing feasibility, and potential environmental impacts would significantly enhance the study's comprehensiveness.

Reply:

Thank you very much for evaluating and recognizing our research. We have added further discussion in terms of the design scalability, manufacturing feasibility, and environmental impact assessment to the revised manuscript (Page 17-18). Please check.

Design scalability is evidenced in **Fig. 2a–d**, where Hammett parameter screening identified BrAce, bearing an electron-withdrawing amide group, as the optimal additive. This establishes an expandable principle for developing next-generation C-Br additives that balance the kinetics and stabilization of I^0/I^+ redox as well as superior anode protection, thereby achieving elevated iodine utilization and prolonged cycle life. The methodology potentially extends to other carbon-halogen systems (e.g., C–Cl, C–F), though their implementation requires nuanced optimization.

Manufacturing feasibility is rooted in cost-effective materials and processes. The electrolyte comprises commercial chemicals (ZnSO_4 salt and BrAce) with raw material expenses substantially below those of alternative four-electron iodine conversion systems (e.g., water-in-salt or eutectic electrolytes), as compared in **Fig. 6f**. Cathode fabrication employs ambient-temperature wet processing with iodine powder and activated carbon (AC), circumventing expensive catalysts or single-walled carbon nanotubes. This approach delivers compelling economic advantages through simplified room-temperature processing and scalable production pathways.

Environmental impact assessment confirms inherent sustainability advantages. Compared to organic electrolytes, the aqueous ZnSO_4 -BrAce system exhibits higher ionic conductivity, lower viscosity, non-flammability, reduced toxicity, and minimal ecological footprint. Critically, BrAce eliminates corrosion issues facing conventional halide salts, as validated by post-cycling Zn anode morphology (**Fig. S31**), Tafel polarization analysis (**Fig. S32**), and stable symmetric/asymmetric cell cycling (**Fig. S33–S34**). The low-concentration electrolyte thus presents negligible environmental hazards while offering significant application potential.

These merits enable carbon-halogen bond-mediated iodine redox as a scalable, manufacturable, and ecologically sound platform for next-generation energy storage. The further discussion has been added to the revised manuscript (Page 17). Please check.

5. Improving the clarity of figures by explicitly labeling schematic pathways, as shown in **Fig. 3f**, would help readers follow the proposed mechanism.

Reply:

Thank you for your nice suggestions. We have provided clearer version of schematic pathways with explicit labeling. Please check.

6. Please correct the inconsistent abbreviation of bromoacetonitrile (BrCN) throughout the figures in the manuscript.

Reply:

Thank you for your nice suggestion. We have corrected all the abbreviation

inconsistency of bromoacetonitrile (BrCN) throughout the figures in the manuscript.

Reviewer #4:

Reply:

Thanks for your valuable comments on our manuscript in conjunction with the other reviewers. Your suggestions will be instrumental in enhancing the quality of our paper.

Response Letter

Dear Reviewers,

Thank you very much for your message pertaining to our submission. We have studied the comments and suggestions. In our point-by-point responses, the Reviewers' comments are presented in black, our responses are provided in **Blue**. The modifications implemented during the revision are indicated in both the revised Manuscript and the Supporting Information, highlighted in **Yellow**. The following is a summary of our revisions and responses to referees.

Reviewer #2:

The authors have addressed my comments successfully.

Reply:

We greatly appreciate your professional evaluation of our research.

Reviewer #3:

Comment on NCOMMS-25-35778A: (Carbon-Halogen Bond Substitution Enables High-Utilization Cascaded Iodine Redox in Noncorrosive Dilute Electrolytes) The authors have made appropriate revisions according to the suggestions. Therefore, I suggest acceptance for publication.

Reply:

Thank you for your positive and professional evaluation of our work.

1. Line 114/532: change “Hametter” to “Hammett”.
2. Lines 230/235: ATR-IR units should be cm^{-1} , not nm; please check for consistency.
3. Replace “electron-drawing” with “electron-withdrawing” in the text and Fig. 1 as they are different in meaning.
4. Supplementary figures: change “figure 50” to “Supplementary Fig. S50” and apply the same style to later figures.

Reply:

Thank you for your suggestions. Issues in the manuscript, including typos, unit inconsistencies, improper word usage, and incorrect abbreviations of image formats, have been corrected. Please review.

Reviewer #4:

Reply:

Thanks for your valuable comments on our manuscript in conjunction with the other reviewers.

Reviewer #1:

I appreciate the authors' efforts to elucidate the proposed C–I bond formation. However, the collected ^1H and ^{13}C NMR spectra at various states of charge and other supplemented experiments do not provide convincing evidence for the claimed C–I⁽⁺⁾–Br covalent bond, which is the central mechanism of this study. If such a covalent bond were indeed formed, the C–I⁽⁺⁾–Br moiety should behave analogously to an alkyl iodine species in organic chemistry. In that case, it would be expected to exhibit characteristic spectroscopic signatures and chemical reactivity distinct from those of simple IBr. The absence of these features casts doubt on the proposed structural assignment.

Reply:

Thank you for your thoughtful questions.

We need to politely reiterate that, given the varying sensitivity of different characterization techniques to the four-electron iodine conversion regulated by BrAce, it is quite challenging to rely on any single method to capture the full picture of the reaction. Instead, comprehensive characterization techniques are essential to obtain a complete evidence chain.

In our investigation, we have employed an integrated array of analytical methods,

including Raman spectroscopy, NMR, FTIR, LC-MS, XPS, and UV-Vis, to confirm the existence of the C-I⁽⁺⁾-Br bond and its marked chemical distinction from conventional I-Br species.

Firstly, we monitored the dynamic changes of the iodine cathode in different electrolytes (BrAce and ZnBr₂) via *in-situ* Raman spectroscopy (**Fig. 3a, b**).

(1) During the charge/discharge process, reversible signals of the I-Br bond at 420-440 cm⁻¹ were clearly identified in both the BrAce and ZnBr₂ systems. This verified that the Br atom of the C-Br bond in the BrAce molecule can bond with I⁺ activated electrochemically.

Meanwhile, in the ZnBr₂ system, distinct characteristic peaks of I₃⁻ and Br₃⁻ were also detected, which is attributed to the slow kinetics and hydrolysis of traditional interhalogen compounds. In contrast, in the BrAce system, only a reversible I⁺ species signal peak was observed at 420-440 cm⁻¹ (within the range of C-1.6 V to D-1.6 V), with no by-product signals detected. This indicates that the I⁺ species formed in BrAce exhibits significant chemical differences from I-Br.

(2) Compared with traditional I-Br, the I⁺ species in the BrAce system shows an obvious redshift, which is consistent with the electron-withdrawing inductive effect (Supplementary Fig. S15). In addition, the intensity of the I⁺ species in the BrAce system is much weaker than that in the ZnBr₂ system. Generally, the intensity of Raman scattering fundamentally depends on the magnitude of polarizability change during molecular vibration. In this work, the electron-withdrawing amide group in BrAce reduces the electron density of adjacent atoms or functional groups, thereby decreasing the dynamic polarizability in the vibration mode and ultimately leading to weakened Raman scattering intensity of C-(I-Br). These results confirmed that the I-Br bond in the BrAce system is not isolated like conventional one, but connected by electron-withdrawing organic group.

(3) Subsequently, we clearly confirmed the existence of C-(I-Br) moiety by LC-MS. As shown in Supplementary **Fig. S21**, a strong signal peak appeared at a m/z of 264.8 when C-1.75 V. This peak is consistent with the molecular weight of the theoretically predicted product NH₂-C(O)-CH₂-I-Br, thus verifying the existence of the

C-(I-Br) structural unit.

Secondly, we further analyzed the bonding mode of C-(I-Br):

(1) Based on Raman spectra of different electrolytes (**Fig. R1**), we identified the characteristic vibrations of SO_4^{2-} and C-Br bond (600 cm^{-1} and 700 cm^{-1} , respectively). As shown in **Fig. R2**, within the *in-situ* Raman testing range of $500\text{--}800\text{ cm}^{-1}$, only the intensity of the SO_4^{2-} characteristic peak changed in the ZnBr_2 system, which may be caused by the variation of the electrolyte environment during charge/discharge. In sharp contrast, within the range of C-1.6 V to D-1.6 V, the BrAce system exhibited reversible split double peaks at around 630 cm^{-1} , which is assigned to the stretching vibration of C-(I-Br) with the I-Br coupling effect.

(2) Meanwhile, the characteristic peak of the C-Br bond disappeared with the formation of C-(I-Br) and gradually recovered after the disappearance of C-(I-Br) (from D-1.6 V to D-0.3 V). This indicates a reversible transformation between the C-Br bond and C-(I-Br). Based on the charge distribution of covalent bonds, the bonding of I^+ enhances the covalent bond, leading to an increase in its force constant and a subsequent rise in vibration frequency (manifested as blueshift). This confirmed that the C is connected with I^+ in the C-(I-Br) moiety, i.e., its presence in the form of C- $\text{I}^{(+)}$ -Br. If bonding occurs in the form of C-Br- $\text{I}^{(+)}$, the characteristic peak of the C-Br bond should undergo a certain degree of blueshift rather than disappearance after I^+ bonding to the end of the C-Br bond. This phenomenon suggests that the C-Br bond breaks and re-bonds in the form of C- $\text{I}^{(+)}$ -Br.

Thirdly, we further investigated the chemical properties of C- $\text{I}^{(+)}$ -Br:

(1) The ^1H NMR spectra of the C- $\text{I}^{(+)}$ -Br sample was compared with those of IAce ($\text{NH}_2\text{-C(O)-CH}_2\text{-I}$) and BrAce. As shown in Supplementary **Fig. S24**, the characteristic signal peak of the α -hydrogen attached to IBr in C- $\text{I}^{(+)}$ -Br appeared between the characteristic peaks of IAce and BrAce. Notably, this signal peak is closer to that of IAce and farther from that of BrAce. This result confirms that the C- $\text{I}^{(+)}$ -Br sample exhibits chemical properties similar to IAce and provides evidence that the alpha-

carbon is bonded with I rather than Br.

(2) Iodine color reaction clearly confirmed the reversible transformation between the C-Br bond and C-I⁽⁺⁾-Br, significant chemical differences between C-I⁽⁺⁾-Br and I-Br, and the similarity of chemical properties of C-I⁽⁺⁾-Br to C-I bond in the IAce. The iodine color reaction is commonly used to determine the presence and characteristics of iodides (*Nature* 1947, 160, 87). Non-polar iodides change color in deuterated chloroform (CDCl₃), which essentially arises from the interaction between the low polarity of CDCl₃ and iodides via van der Waals forces, and this weak interaction barely alters the electronic structure of iodides. As shown in **Fig. R3a**, BrAce and IAce appear colorless and magenta in CDCl₃, respectively, while I⁻ (from ZnI₂), I₂, and I-Br appear colorless, purple-black, and yellow-brown in CDCl₃, respectively. When the products of the iodine cathode at different potentials (with the same iodine content) were dissolved in CDCl₃, it was found that C-Br^{••}I⁰ (C-1.3 V) had the same color as BrAce, while the colors of I^{••}C^{••}Br (C-1.5 V) and C-I-Br (C-1.75 V, D-1.6 V) gradually approached that of IAce (Fig. R3b).

Finally, we assembled a Zn||I₂ battery using IAce (0.4 M, ~5 mgI cm⁻²) as the iodine source, ZnBr₂ (0.35 M) as the initiator, and carbon cloth as the cathode current collector. The results showed that within the voltage window of 0.3–1.75 V, distinct I⁻/I⁰/I⁺ redox couple characteristic peaks appeared (**Fig. R4**). In addition, the battery showed no obvious capacity decay during 500 cycles, indicating that a stable C-I⁽⁺⁾-Br structure was also formed in the system. This result further verifies the reliability of the above conclusions.

This relevant data and explanation have been added to the revised Manuscript (Pages 7-8, 10-11) and Supporting Information (Pages 11-13). Please see the highlighted parts.

Fig. R1 Raman spectra of ZSO, ZSO+ZnBr₂, ZSO+Ace, and ZSO+BrAce electrolytes.

Fig. R2 (a) Typical charge/discharge curves and corresponding *in-situ* Raman spectra of cathodes in (b) 0.35 M ZnBr₂ and (c) 0.7 M BrAce electrolytes.

Fig. R3 (a) Photographs for BrAce, IAce, and different iodides dissolved in CDCl_3 . (b) Photographs for the products of the iodine cathode at different potentials dissolved in CDCl_3 .

Fig. R4 (a) GCD curves, (b) corresponding differential charge-discharge curves, and (c) corresponding long-term cycling performance of $\text{Zn}||\text{I}_2$ batteries with IAce+ZnBr₂ electrolyte.

2. Furthermore, if the “electron-withdrawing amide group of BrAce diminishes electron density in adjacent atoms,” then the adjacent I^+ center should be less polarized. A less polarized I^+ would be expected to exhibit a lower valence state compared to IBr, which contradicts both the XPS data and the ^{13}C NMR results. In addition, such a less polarized I^+ in the $\text{C}-\text{I}^{(+)}-\text{Br}$ structure should yield a lower discharge voltage than

conventional IBr, which is not observed.

Reply:

Thank you for your thoughtful questions.

The influence of microscopic electron cloud density on the macroscopic valence state of a compound depends on the bonding type of compounds (*Angew. Chem. Int. Ed.* 2020, 59, 984-1001). Both C-X bonds and I-X compounds exist as covalent bonds, where atoms form chemical bonds through shared electron pairs. The electron pairs are "shared" between atoms, with no obvious electron transfer. Although the electron-withdrawing effect can alter the distribution of local electron cloud density, it cannot break the electron-sharing characteristic of covalent bonds. Therefore, it does not cause atoms to actually gain or lose electrons, and thus does not directly change the valence state.

Usually, I^+ species exist in the form of interhalogen compounds (e.g., IBr, ICl). In concentrated electrolytes containing halogen anions ($X^- = Br^-, Cl^-$), the I-X can further form chain-like molecular structures (e.g., $[\cdots I-X\cdots I-X\cdots]^n$) or anion clusters (e.g., IX_2^- , IX_3^-). **Fig. R5** presents the previously reported multi-electron iodine conversion via I-Cl or I-Br bonds ($I/I-Cl/ICl_2^-$, *Nat. Commun.* 2023, 14, 1856; $I/I-Br_2^-/IBr_3^-$, *Adv. Mater.* 2025, <https://doi.org/10.1002/adma.202415979>). It can be seen that when the I^+ center is bonded to multiple electron-withdrawing groups (Cl^-, Br^-), there is no significant shift in the XPS spectra of the I^+ species. In our work, in the I 3d XPS spectra (Supplementary Fig. S18), the characteristic peak of C-I-Br is slightly lower than that of I-Br (631.5 eV vs. 631.8 eV). Notably, it is closer to the C-I bond position (631.4 eV). This result is consistent with the literature above. This also provides evidence that C-I-Br exhibits similar properties to alkyl iodide substances.

The discharge voltage and polarization of a battery also need to consider multiple factors, such as product stability and anode corrosion. When the charging products undergo hydrolysis, the battery polarization effect can be intensified, leading to a decrease in discharge voltage (*Nat. Energy* 2024, 9, 714-724). This is because hydrolysis results in the formation of various by-products, which in turn causes the

mismatch between ion transport and the demand of electrode reactions. As a consequence, ohmic polarization, electrochemical polarization, and concentration polarization are all significantly enhanced, making the actual discharge voltage deviate from the equilibrium potential. The relevant data and explanation are as follows:

1) Combining with *in-situ* Raman spectra, UV-Vis spectra, and hydrolysis energy barrier calculations, it can be concluded that severe hydrolysis and the shuttling effect of I_3^- and Br_3^- will significantly disrupt the I-Br reaction pathway. In contrast, C-I-Br exhibits a higher hydrolysis energy barrier and better stability.

2) As shown in Supplementary **Fig. S47**, compared with BrAce, $ZnBr_2$ exhibits stronger corrosivity toward the Zn anode, which hinders ion transport at the anode interface. Therefore, the C-I-Br exhibits higher capacity than the I-Br, with a more complete high-plateau reaction, higher discharge voltage, and lower polarization. Meanwhile, the multi-electron conversion systems of $I^-/I^0/I^+$ and Br^-/Br^0 reported in relevant literature (*Adv. Mater.* 2024, 36, 2401924; *Adv. Mater.* 2025, <https://doi.org/10.1002/adma.202415979>) also indicates that after activation of the Br^-/Br^0 redox couple, the discharge plateau of $I^-/I^0/I^+$ decreases to a certain extent due to the aggravation of side reactions (**Fig. R6**).

This relevant data and explanation have been added to the revised Manuscript (Page 8). Please see the highlighted parts.

Figure Redacted

Fig. R5 Multi-electron conversion systems reported in relevant literature: (a) $I^-/I-Cl/I-Cl_2^-$ (*Nat. Commun.* 2023, 14, 1856); (b) $I^-/IBr_2^-/IBr_3^-$ (*Adv. Mater.* 2025, <https://doi.org/10.1002/adma.202415979>).

Figure Redacted

Fig. R6 Multi-electron conversion systems reported in relevant literature: (a) $I^+/I_2/I^+$ (*Adv. Mater.* 2024, 36, 2401924); (b) Br^-/Br^0 (*Adv. Mater.* 2025, <https://doi.org/10.1002/adma.202415979>).

3. The self-discharge tests also raise concerns: the I^+/I_2 plateau is nearly absent in the charge–discharge curves, suggesting that BrAce electrolytes fail to stabilize I^+ effectively. Moreover, under high iodine loading, the capacity contribution from the

I^+/I^0 redox process is minimal, indicating that the $I^0 \rightarrow I^+$ oxidation step may be kinetically hindered.

Reply:

Thank you for your insightful suggestions.

It has been acknowledged that in terms of self-discharge performance, $4e^- Zn||I_2$ batteries do not show obvious advantages compared with traditional $2e^- Zn||I_2$ batteries. This is a long-standing challenge within the research field of $4e^- Zn||I_2$ batteries.

In halogen anion ($X^-=Br^-, Cl^-$)-dilute electrolytes, the generated interhalogen compounds are prone to hydrolysis, due to the presence of active axial δ -hole of interhalogen bonds. The hydrolysis of I^+ significantly impairs its self-discharge resistance. Nevertheless, the BrAce system still exhibits remarkable advantages over the traditional $ZnBr_2$ system: after 12 h of storage, its capacity retention rate is much higher than that of $ZnBr_2$ (77.44% vs. 42.6%).

Meanwhile, we have also compared the self-discharge performances reported in the state-of-the-art relevant literature on $4e^- Zn||I_2$ batteries via other modulation strategies (e.g., high-concentration salt electrolyte, *Nat. Commun.* 2021, 12, 170; support adsorption-catalysis, *Sci. Bull.* 2024, 69, 1674-1685; Organic halide additive, *J. Am. Chem. Soc.* 2025, 147, 19, 16350-16361; eutectic electrolytes, *Adv. Energy Mater.* 2025, <https://doi.org/10.1002/aenm.202501460>, and N-I coordination, *Energy Environ. Sci.* 2023, 16, 4502-4510). As shown in the **Fig. R7** and **Table R1**, it can be seen that compared with these strategies, the self-discharge performance regulated by BrAce remains at a relatively high level. Therefore, regulating $4e^-$ iodine conversion via carbon-halogen bonds is a feasible strategy. In the future, a combination of multiple regulation approaches may be adopted to better address the severe self-discharge issue caused by I^+ hydrolysis.

Notably, compared with existing strategies, carbon-halogen bond regulation exhibits distinct kinetic advantages. As shown in **Fig. R8**, existing strategies can barely achieve charge-discharge tests under high rates. When the current density exceeds $5 A g^{-1}$, the capacity contributed by the I^0/I^+ conversion decreases sharply; even though

some literature has conducted tests at current densities above 20 A g^{-1} , they can hardly maintain high-plateau conversion ($1\text{--}2 \text{ mg cm}^{-2}$). In contrast, the BrAce system enables ultrahigh-rate operation ($1\text{--}60 \text{ A g}^{-1}$) with a load of 4 mg cm^{-2} while sustaining stable dual-plateau conversion (Fig. R10). The excellent rate capability indicates that it possesses fast reaction kinetics. Meanwhile, under high-loading conditions, the existing systems only show a very low proportion of capacity contributed by high-plateau conversion when the areal capacity reaches 2 mA h cm^{-2} (Fig. R9); in contrast, the BrAce system can still maintain stable dual-plateau conversion at a high areal capacity of $3.85 \text{ mA h cm}^{-2}$ and a large current density of 47.5 mA cm^{-2} .

All these results demonstrate that the carbon-halogen bond regulation strategy reduces the reaction barrier of the I^0/I^+ conversion and enhances its conversion kinetics.

Figure Redacted

Fig. R7 Different strategies for regulating the self-discharge performance of 4e^- iodine conversion. (a) High-concentration salt electrolyte (*Nat. Commun.* 2021, 12, 170), (b) support adsorption-catalysis (*Sci. Bull.* 2024, 69, 1674-1685), (c) organic halide additive (*J. Am. Chem. Soc.* 2025, 147, 19, 16350-16361), (d) eutectic electrolyte (*Adv. Energy Mater.* 2025, <https://doi.org/10.1002/aenm.202501460>), and (e) N-I coordination (*Energy Environ. Sci.* 2023, 16, 4502-4510).

Figure Redacted

Fig. R8 Different strategies for regulating the GCD curves of $4e^-$ iodine conversion. (a) High-concentration salt electrolyte (*Angew. Chem. Int. Ed.* 2025, <https://doi.org/10.1002/anie.202515633>), (b) Eutectic electrolyte (*Angew. Chem. Int. Ed.* 2025, <https://doi.org/10.1002/anie.202514375>), (c) Organic halide additive (*Angew. Chem. Int. Ed.* 2025, <https://doi.org/10.1002/anie.202513747>), (d) Support adsorption-catalysis (*Adv. Mater.* 2024, 36, 2312246), and (e) N-I coordination (*J. Am. Chem. Soc.* 2025, 147, 30, 26889 – 26897).

Figure Redacted

Fig. R9 GCD curves of $4e^-$ iodine conversion under (a) high loading (*Angew. Chem. Int. Ed.* 2024, 63, e202404784) and (b) pouch cell (*Adv. Mater.* 2025, <https://doi.org/10.1002/adma.202514117>) conditions in relevant literature.

Table R1. Different self-discharge performance of 4e⁻ iodine conversion in literatures.

Different strategies	Rest time	CE
High-concentration electrolyte	salt 12 h	80.9%
Support catalysis	adsorption- 6 h	70.5%
Organic halide additive	6 h	82.4%
Eutectic electrolyte	12 h	63.67%
N-I coordination	3 h	93.5%
This work	12 h	77.44%

4. The comparison of voltage polarization reported in Figure 2e is not fair.

Reply:

Thanks for the thoughtful question.

When we compared the discharge curves at different rates individually (**Fig. R10**), we can see that at current densities ranging from 1 to 60 A g⁻¹, the discharge plateaus corresponding to the I⁻/I⁰ and I⁰/I⁺ redox couples in the BrAce-containing system are well-maintained. Additionally, the discharge capacity is much higher than that of the ZnBr₂ system, and the voltage polarization is significantly lower than that of the ZnBr₂ system.

This relevant data and explanation have been added to the revised Manuscript (Page 7) and Supporting Information (Page 8). Please see the highlighted parts.

Fig. R10 Discharge curves of Zn||I₂ batteries at ultrahigh rates of 1-60 A g⁻¹ with BrAce and ZnBr₂ electrolytes.

Given these issues, the mechanistic interpretation of iodine transformations remains insufficiently justified. I regret that I cannot provide a positive assessment of this work again.

Reply:

We greatly appreciate your concerns about our research and your professional comments. We have provided the above response to address your concerns.